# Improving Backward Conformal Prediction via Non-Conformity Score Transformation

Junxian Liu [1,2]   Hao Zeng [1]   Hongxin Wei [1]

## Abstract

Conformal Prediction (CP) provides a statistical framework for uncertainty quantification that constructs prediction sets with coverage guarantees. While CP yields uncontrolled prediction set sizes, Backward Conformal Prediction (BCP) inverts this paradigm by enforcing a predefined upper bound on set size and estimating the resulting coverage guarantee. However, the looseness induced by Markov's inequality within the BCP framework causes a significant gap between the estimated coverage bound and the empirical coverage. In this work, we introduce ST-BCP, a novel method that introduces a data-dependent transformation of nonconformity scores to narrow the coverage gap. In particular, we develop a computable transformation and prove that it outperforms the baseline identity transformation. Extensive experiments demonstrate the effectiveness of our method, reducing the average coverage gap from 4.20% to 1.12% on common benchmarks.

## 1. Introduction

Machine learning models are increasingly deployed in high-risk domains such as medical diagnosis (Caruana et al., 2015), and autonomous driving (Bojarski et al., 2016). In these settings, incorrect predictions can lead to severe consequences, underscoring the critical need for reliable uncertainty quantification. Conformal Prediction (CP) (Vovk et al., 2005; Angelopoulos et al., 2024) provides a distribution-free framework that converts point predictions into prediction sets containing the true label with a user-specified probability. This framework is particularly well-suited for modern deep learning models, as it ensures

rigorous statistical coverage without requiring model retraining or distributional assumptions (Fontana et al., 2023).

Despite its theoretical advantage, Conformal Prediction often yields excessively large prediction sets (Xu et al., 2025; Shi et al., 2024), thereby diminishing its practical utility for tasks that require informative, concise outputs. To address this, Backward Conformal Prediction (BCP) (Gauthier et al., 2025b) inverts this paradigm by pre-defining an upper bound on prediction set size and estimating the corresponding coverage bound. However, the looseness introduced by applying Markov's inequality to e-variables within the BCP framework yields excessively conservative coverage guarantees, resulting in a significant gap between empirical coverage and the estimated coverage bound. This motivates us to tighten the coverage bound by mitigating the looseness of Markov's inequality.

In this work, we propose ST-BCP, a novel method that utilizes a data-dependent transformation of non-conformity scores to narrow the coverage gap. Our key idea is to reshape the distribution of non-conformity scores to mitigate the looseness of Markov's inequality, thereby tightening the coverage bound. A central challenge in such data-dependent transformations is preserving exchangeability, the mathematical foundation of CP validity. We resolve this by employing a symmetric parameterization strategy: defining transformation parameters by combining the leave-one-out calibration set with a pseudo-test point.

Furthermore, we ensure that the coverage bound can still be consistently estimated under mild regularity assumptions. Theoretically, we derive the optimal transformation under a monotonicity constraint. Since this theoretical optimum relies on unknown conditional distributions, we provide a computable approximation that closely mimics the optimal behavior. Notably, we provide a formal proof demonstrating that this approximation yields superior performance relative to the baseline identity transformation.

ST-BCP delivers a significantly tighter lower bound on coverage compared to BCP, thereby reducing the conservatism inherent in coverage estimation. This improvement is practically valuable in downstream applications, as excessively conservative coverage bounds may prompt unwarranted human intervention. Consider a medical diagnostic system

---

[1]Department of Statistics and Data Science, Southern University of Science and Technology [2]College of Mathematics and Statistics, Chongqing University. Correspondence to: Hongxin Wei <weihx@sustech.edu.cn>.

*Proceedings of the $43^{rd}$ International Conference on Machine Learning*, Seoul, South Korea. PMLR 306, 2026. Copyright 2026 by the author(s).

where routine cases are automated if the estimated coverage lower bound exceeds a safety threshold; otherwise, it conservatively defers to a doctor. With standard BCP, the estimated coverage lower bound may fall severely below the threshold even if the true coverage is safe. This over-conservatism forces safe cases into manual review. By applying ST-BCP, the estimated coverage lower bound successfully exceeds the required threshold, accurately reflecting the system's true reliability and significantly reducing unnecessary human intervention.

Extensive experiments across CIFAR-10, CIFAR-100, and Tiny-ImageNet demonstrate the superiority of our proposed score transformation over the original BCP framework. First, our method drastically narrows the coverage gap between the leave-one-out (LOO) estimator and empirical coverage. For instance, using a ResNet-50 model on CIFAR-10 with the size constraint rule of $\mathcal{T} = 2$, our transformation reduces the coverage gap (GAP) from 5.38% to 0.72%—a nearly seven-fold improvement over the baseline identity transformation. Beyond the constant size constraint rule, our approach also yields a significantly tighter bound under a complex feature- and calibration-set-dependent size constraint rule, reducing GAP from 3.57% to 1.27%. Furthermore, our method consistently achieves lower MSE and standard deviation as the calibration set size $n$ increases. This confirms that our approach improves estimation precision while maintaining asymptotic efficiency comparable to the original BCP. Code is available at https://anonymous.4open.science/r/ST-BCP-F2487B.

We summarize our primary contributions as follows:

1. We formalize the theoretical conditions under which data-dependent transformations preserve score exchangeability and establish the criteria for the leave-one-out estimator to remain a consistent estimator of the coverage bound.

2. We analytically derive the optimal transformation under a monotonicity constraint and develop a computationally tractable approximation that enables direct implementation without requiring inaccessible distributional information.

3. Through extensive empirical validation, we demonstrate that our proposed method significantly narrows the gap between empirical coverage and the estimated coverage bound, while maintaining a convergence rate comparable to the baseline identity transformation.

## 2. Preliminaries

In this work, we focus on classification tasks. Let $\mathcal{X}$ be the feature space and $\mathcal{Y}$ be the label space with cardinality $|\mathcal{Y}|$. Given a pre-trained model $f$, let $D_{n+1} = \{(X_i, Y_i)\}_{i=1}^n$

denote the calibration set. We further let $(X_{n+1}, Y_{n+1})$ be the test point, where $X_{n+1}$ is observed at inference time, while the true label $Y_{n+1}$ remains unknown.

**Conformal Prediction**: Conformal Prediction (CP) provides a versatile and distribution-free framework for uncertainty quantification. The core objective of CP is to construct a prediction set $C^\alpha(X_{n+1})$ that ensures a marginal coverage guarantee at a user-specified error level $\alpha \in (0, 1)$:

$$\mathbb{P}(Y_{n+1} \in C^\alpha(X_{n+1})) \geq 1 - \alpha \tag{1}$$

To implement this framework, CP relies on a non-conformity score function $S : \mathcal{X} \times \mathcal{Y} \to \mathbb{R}^+$, which quantifies the discrepancy between an observation $(X, Y)$ and the model $f$. By using these scores calculated over the calibration set $D_{cal}$ and the test instance, CP constructs a statistical event involving the true label of the test point. The prediction set is then formed by traversing the label space and including all candidate labels $y \in \mathcal{Y}$ that satisfy this event, offering a rigorous and formalized assurance for predictive performance.

**Conformal e-Prediction**: Conformal e-Prediction (Vovk, 2025) represents a specific implementation of CP that utilizes e-variables and Markov's inequality to derive coverage bounds. An e-variable is defined as a non-negative random variable with an expectation of at most 1. In this work, we employ the following construction for the e-variable at the test point (Balinsky & Balinsky, 2024; Wang & Ramdas, 2022; Koning, 2023):

$$E^{n+1}(Y_{n+1}) = \frac{(n+1)S(X_{n+1}, Y_{n+1})}{\sum_{i=1}^n S_i + S(X_{n+1}, Y_{n+1})} \tag{2}$$

Under the condition that the scores $S_1, \ldots, S_{n+1}$ are exchangeable, it can be shown that $\mathbb{E}[E^{n+1}(Y_{n+1})] = 1$. By applying Markov's inequality, we obtain the statistical event $E^{n+1}(Y_{n+1}) < 1/\alpha$ with a guaranteed probabilistic lower bound:

$$\mathbb{P}\left(E^{n+1}(Y_{n+1}) < \frac{1}{\alpha}\right) \geq 1 - \alpha\mathbb{E}[E^{n+1}(Y_{n+1})] = 1 - \alpha \tag{3}$$

Accordingly, the prediction set is defined as

$$C^\alpha(X_{n+1}) = \{y \in \mathcal{Y} : E^{n+1}(y) < \frac{1}{\alpha}\} \tag{4}$$

This framework provides the necessary mathematical foundation for Backward Conformal Prediction (BCP), as it allows for adaptive (data-dependent) miscoverage level while strictly maintaining post-hoc validity (Gauthier et al., 2025c). This flexibility is pivotal for inverting the standard conformal paradigm to enforce strict constraints on prediction set size.

**Backward Conformal Prediction:** Backward Conformal Prediction (BCP) (Gauthier et al., 2025b) inverts the standard CP paradigm by pre-defining an upper bound on the

prediction set size to ensure practical utility. The size constraint is represented by a function:

$$\mathcal{T} : \mathcal{D} \times \mathcal{X} \to \{1, \ldots, |\mathcal{Y}|\} \quad (5)$$

BCP requires the event $|C^\alpha(X_{n+1})| \leq \mathcal{T}_{n+1}$ to hold, where $\mathcal{T}(D_{n+1}, X_{n+1}) = \mathcal{T}_{n+1}$. This is achieved by introducing a data-dependent miscoverage level $\tilde{\alpha}_{n+1}$, defined as the smallest $\alpha$ that satisfies the size constraint. Since the prediction sets are constructed as open sets, the existence of $\tilde{\alpha}_{n+1}$ is guaranteed.

$$\tilde{\alpha}_{n+1} = \inf\{\alpha \in (0,1) : |C^\alpha(X_{n+1})| \leq \mathcal{T}_{n+1}\} \quad (6)$$

Based on the post-hoc validity of Conformal e-Prediction and a first-order Taylor approximation (Gauthier et al., 2025c), BCP provides the following coverage guarantee:

$$\mathbb{P}(Y_{n+1} \in C^{\tilde{\alpha}_{n+1}}(X_{n+1})) \geq 1 - \mathbb{E}[\tilde{\alpha}_{n+1}] \quad (7)$$

To implement this guarantee in practice, the expectation $\mathbb{E}[\tilde{\alpha}_{n+1}]$ is estimated via the leave-one-out (LOO) estimator $\hat{\alpha}^{LOO} = \frac{1}{n}\sum_{i=1}^{n}\tilde{\alpha}_i$. Here, each pseudo miscoverage level $\tilde{\alpha}_i$ is calculated symmetrically to Eq. (6) by treating the $i$-th calibration point as a pseudo-test point. Under mild regularity assumptions, $\hat{\alpha}^{LOO}$ is a consistent estimator for the theoretical expectation:

$$|\hat{\alpha}^{LOO} - \mathbb{E}[\tilde{\alpha}_{n+1}]| = O_P\left(\frac{1}{\sqrt{n}}\right) \quad (8)$$

A critical limitation of BCP is the significant gap between the estimated coverage bound and the empirical coverage. This coverage gap primarily stems from a fundamental mismatch between the inherent distribution of non-conformity scores and the statistical tool utilized to provide the coverage guarantee. Specifically, the coverage bound in Eq. (7) is derived via Markov's inequality, which only utilizes the expectation (i.e., first-order information) of the miscoverage level and remains insensitive to the specific shape or tail behavior of the score distribution. Consequently, the original scores often fail to effectively fit this conservative mathematical constraint, leading to an excessively loose bound. This mismatch motivates our proposed method, which introduces a score transformation to reshape the score distribution.

## 3. Method

In the previous analysis, the excessive coverage gap was due to the excessive looseness caused by the inequality; in other words, it was the mismatch between the non-conformity score distribution and Markov's inequality. Our key idea is to find a suitable data-dependent transformation to reshape the distribution to adapt to Markov's inequality while maintaining the validity of BCP.

Since the data-dependent transformations introduce additional theoretical complexity, we first formalize the ST-BCP

formulation under a broad class of general transformations. In order to find the optimal candidate, we directly cast the minimization of the coverage gap as a functional optimization problem. This approach yields a transformation with superior performance over the identity transformation (original BCP), effectively tightening the bound loosened by Markov's inequality.

### 3.1. The General Formulation of ST-BCP

**Score Transformation.** To minimize the coverage gap, we introduce a non-negative, data-dependent transformation $h(s; D, X)$. Here, $s$ represents the non-conformity score, while $D$ and $X$ serve as parameters incorporating dataset and feature information. This design is intrinsically coupled with the size constraint rule $\mathcal{T}(D, X)$, thereby enabling the transformation to exploit the information of $\mathcal{T}$.

**Symmetric Parameterization.** To preserve exchangeability—the mathematical foundation of CP validity—we employ a symmetric parameterization strategy that incorporates candidate labels to handle the unknown true label $Y_{n+1}$. For a candidate label $y \in \mathcal{Y}$, the transformed scores are constructed as follows:

- **Calibration Scores:** $h_i^y = h(S_i; D_i^y, X_i)$, where $D_i^y = D_i \cup \{(X_{n+1}, y)\} \quad D_i = D_{n+1} \setminus \{(X_i, Y_i)\}$

- **Test Score:** $h_{n+1}(y) = h(S(X_{n+1}, y); D_{n+1}, X_{n+1})$

Under this construction, consider the case when $y = Y_{n+1}$: the dataset parameter for each term comprises all data points excluding the instance itself, while the feature parameter corresponds to the instance's own feature. This structural uniformity ensures that the sequence $\{h_1^{Y_{n+1}}, \ldots, h_{n+1}(Y_{n+1})\}$ becomes mathematically symmetric, thereby strictly preserving exchangeability.

**Prediction Set and Miscoverage Level.** Following the BCP framework, the transformed e-variable $E^{n+1}(y, h)$ is defined as:

$$E^{n+1}(y, h) = \frac{(n+1)h_{n+1}(y)}{\sum_{i=1}^{n} h_i^y + h_{n+1}(y)} \quad (9)$$

The prediction set $C_h^\alpha(X_{n+1})$ is constructed as follows:

$$C_h^\alpha(X_{n+1}) = \{y \in \mathcal{Y} : E^{n+1}(y, h) < 1/\alpha\} \quad (10)$$

Given the size constraint $\mathcal{T}_{n+1} = \mathcal{T}(D_{n+1}, X_{n+1})$, the data-dependent miscoverage level $\tilde{\alpha}_{n+1}(h)$ is the minimum $\alpha$ satisfying this size requirement:

$$\tilde{\alpha}_{n+1}(h) = \inf\{\alpha \in (0,1) : |C_h^\alpha(X_{n+1})| \leq \mathcal{T}_{n+1}\} \quad (11)$$

Since $E^{n+1}(Y_{n+1}, h)$ remains a valid e-variable under exchangeability, we utilize a first-order Taylor approximation

to establish the coverage guarantee. The impact of score transformation on the reliability of this approximation is analyzed in Appendix E.

$$\mathbb{P}(Y_{n+1} \in C_h^{\tilde{\alpha}_{n+1}(h)}(X_{n+1})) \geq 1 - \mathbb{E}[\tilde{\alpha}_{n+1}(h)] \quad (12)$$

**LOO Estimator.** The primary objective of the LOO estimator is to estimate $\mathbb{E}[\tilde{\alpha}_{n+1}(h)]$ using only the calibration set and test feature, thereby bypassing the unknown true label of the test point. To maintain structural symmetry with the original transformation while ensuring computability, the transformation is adapted by treating each calibration point as a pseudo-test point. For each $i \in \{1, \ldots, n\}$, we define the transformed scores as follows:

- **Pseudo Calibration Scores:** $h_j = h(S_j; D_j, X_j)$.

- **Pseudo Test Score:** $h_i(y) = h(S(X_i, y); D_i, X_i)$.

The pseudo e-variable $E^i(y, h)$ is constructed as follows:

$$E^i(y, h) = \frac{n h_i(y)}{\sum_{j \neq i}^n h_j + h_i(y)} \quad (13)$$

Let $\mathcal{T}_i = \mathcal{T}(D_i, X_i)$ and $C_h^\alpha(X_i) = \{y \in \mathcal{Y} : E^i(y, h) < 1/\alpha\}$. The corresponding pseudo miscoverage levels $\tilde{\alpha}_i(h)$ and the LOO estimator $\hat{\alpha}^{LOO}(h)$ are defined as:

$$\tilde{\alpha}_i(h) = \inf\{\alpha \in (0, 1) : |C_h^\alpha(X_i)| \leq \mathcal{T}_i\} \quad (14)$$

$$\hat{\alpha}^{LOO}(h) = \frac{1}{n}\sum_{i=1}^n \tilde{\alpha}_i(h) \quad (15)$$

The introduction of the data-dependent transformation $h(s; D, X)$ inevitably creates intricate statistical dependencies among the non-conformity scores, potentially complicating the estimation process. However, the following theorem rigorously establishes the consistency of our estimator despite these complex couplings:

**Theorem 3.1.** *Under appropriate regularity conditions, the leave-one-out estimator satisfies:*

$$|\hat{\alpha}^{LOO}(h) - \mathbb{E}[\tilde{\alpha}_{n+1}(h)]| = O_P\left(\frac{1}{\sqrt{n}}\right) \quad (16)$$

The proof and detailed conditions of Theorem 3.1 are provided in Appendix F. Theorem 3.1 serves as the theoretical bridge between the computable LOO estimator and the uncomputable coverage bound. It guarantees that despite the scores being reshaped by contextual information, the estimator maintains an $O_P(1/\sqrt{n})$ convergence rate comparable to the original BCP framework. This result confirms the practical operationalization of ST-BCP, which we can reliably employ $\hat{\alpha}^{LOO}(h)$ as the estimated coverage bound.

### 3.2. Optimization of Score Transformation

**Coverage Gap Decomposition.** To understand the effect of score transformation, we first decompose the overall coverage gap:

$$\text{GAP}(h) = \left|\hat{\alpha}^{LOO}(h) - \mathbb{P}\left(Y_{n+1} \notin C_h^{\tilde{\alpha}_{n+1}(h)}(X_{n+1})\right)\right| \quad (17)$$

Using the triangle inequality, we obtain

$$\text{GAP}(h) \leq \underbrace{\left|\hat{\alpha}^{LOO}(h) - \mathbb{E}[\tilde{\alpha}_{n+1}(h)]\right|}_{\text{estimation error}} +$$
$$\underbrace{\left|\mathbb{E}[\tilde{\alpha}_{n+1}(h)] - \mathbb{P}\left(Y_{n+1} \notin C_h^{\tilde{\alpha}_{n+1}(h)}(X_{n+1})\right)\right|}_{\text{looseness and approximation error}}$$

The first term corresponds to the estimation error of the LOO estimator. Under the conditions of Theorem 3.1, this term converges to zero at rate $O_P(n^{-1/2})$. Therefore, the score transformation has limited influence on this component, and we do not optimize it explicitly. The second term combines both the first-order Taylor approximation error and the looseness introduced by Markov's inequality. When the first-order Taylor approximation is accurate, Eq. (7) implies that the absolute value can be removed.

Under this regime, reducing the looseness induced by Markov's inequality becomes approximately equivalent to reducing the overall coverage gap. Consequently, we focus on minimizing this looseness. We formulate this as a functional optimization problem where the score transformation $h$ serves as the optimization variable:

$$\min_h : \mathbb{E}[\tilde{\alpha}_{n+1}(h)] - \mathbb{P}\left(Y_{n+1} \notin C_h^{\tilde{\alpha}_{n+1}(h)}(X_{n+1})\right)$$

**Optimization Objective Simplification.** Since the underlying distribution of $(X, Y)$ is unknown, $\mathbb{P}(Y_{n+1} \in C_h^{\tilde{\alpha}_{n+1}(h)}(X_{n+1}))$ and $\mathbb{E}[\tilde{\alpha}_{n+1}(h)]$ cannot be expressed as functionals that are easily optimized without imposing additional constraints on the transformation. By requiring $h(s; D, X)$ to be strictly monotonic in $s$, we can leverage its inverse to significantly simplify the derivation. Moreover, to ensure both $h(s; D, X) \geq 0$ and the preservation of the original label score ordering, $h(s; D, X)$ must be strictly monotonically increasing with respect to $s$. Let $\mathcal{H}$ denote the class of functions $h(s; D, X)$ that are strictly monotonically increasing in $s$.

**Theorem 3.2.** *Under appropriate conditions, $\forall h^1, h^2 \in \mathcal{H}$, we have:*

$$C_{h^1}^{\tilde{\alpha}_{n+1}(h^1)}(X_{n+1}) = C_{h^2}^{\tilde{\alpha}_{n+1}(h^2)}(X_{n+1}) \quad (18)$$

*Remark* 3.3. If $\frac{1}{n}\sum_{i=1}^n h_i = \frac{1}{n}\sum_{i=1}^n h_i^y$ always holds true, this conclusion also holds for all $n \in \mathbb{N}^+$.

We provide the detailed conditions and the formal proof of Theorem 3.2 in Appendix G. Crucially, this theorem establishes a fundamental invariance property: it guarantees that, under appropriate conditions, any transformation within the class $\mathcal{H}$ yields prediction sets identical to those obtained under the original scale. Notably, as highlighted in Remark 3.3, these conditions are automatically satisfied when the transformation is independent of the dataset parameter $D$. This invariance effectively simplifies the complex objective of narrowing the coverage gap. Consequently, we focus solely on the optimization problem: minimizing the expected miscoverage level $\mathbb{E}[\tilde{\alpha}_{n+1}(h)]$.

**Corollary 3.4.** *Under the same conditions used in theorem 3.2, we can draw the following corollary:*

$$\tilde{\alpha}_{n+1}(h) = \frac{1}{n+1}\left(\frac{\sum_{i=1}^{n} h_i}{h(w_{n+1}; D_{n+1}, X_{n+1})} + 1\right) \quad (19)$$

$$\tilde{\alpha}_i(h) = \frac{1}{n}\left(\frac{\sum_{j\neq i}^{n} h_j}{h(w_i; D_i, X_i)} + 1\right) \quad i = 1, \ldots, n \quad (20)$$

*where,*

$$w(D, X) = \sup\{l > 0 : |\{y : S(X, y)\}| \leq \mathcal{T}(D, X)\}$$

$$w_i = w(D_i, X_i) \quad i = 1, \cdots, n+1$$

This threshold function $w(D, X)$ effectively maps the discrete cardinality constraint onto the continuous score space, thereby converting the miscoverage level from its implicit infimum form into an analytically tractable closed-form expression. Especially in certain cases, $w(D, X)$ itself admits a simple closed-form solution, as detailed in Appendix D. Using Corollary 3.4, the optimization objective can be simplified to a ratio-type functional.

$$\min_{h\in\mathcal{H}} : \mathbb{E}[\tilde{\alpha}_{n+1}(h)] \Leftrightarrow \min_{h\in\mathcal{H}} : \mathbb{E}\left[\frac{h_i}{h(w_{n+1}; D_{n+1}, X_{n+1})}\right]$$

**Optimal Transformation.** While the strict monotonicity of $h(s; D, X)$ is necessary to transform the coverage gap into this specific form, it presents a theoretical challenge: under some function space topologies, the set of strictly monotonically increasing functions is typically an open set. This openness may lead to the non-existence of an optimal solution within $\mathcal{H}$. To ensure a well-defined optimizer, we consider the set of monotonically non-decreasing functions, which can be viewed as the closure of the strictly increasing set. This closed set, combined with the fact that the objective functional is bounded from below, guarantees the existence of an optimal solution. Therefore, we expand our search space to $\bar{\mathcal{H}}$, defined as the class of functions $h(s; D, X)$ that are monotonically non-decreasing with respect to $s$.

**Theorem 3.5.** *Under appropriate conditions, we have:*

$$h^{opt}(s; D, X) = \frac{a\mathbb{I}(s \geq w(D, X))}{\sqrt{\mathbb{P}(S(X, Y) \geq w(D, X)|D, X)}}$$

$$= \underset{h\in\mathcal{H}}{argmin} : \mathbb{E}\left[\frac{h_i}{h(w_{n+1}; D_{n+1}, X_{n+1})}\right]$$

*where $a$ is an arbitrary positive constant.*

The detailed conditions and the proof of Theorem 3.5 are provided in Appendix H. Given that $h(s; D, X)$ is non-monotonic decreasing with respect to $s$, the structural search space is constrained such that the optimal $h$ must be a step function. For fixed $X$ and $D$, the transformation is zero below the threshold $w(D, X)$ and a positive constant thereafter. Then the optimal solution is formally derived using the variational method.

**Improvement Operator.** Although this transformation is theoretically optimal, its direct application is hindered by the requirement for unknown conditional probabilities. The derivation of this optimum inspires an interpretation of the procedure as a refinement process induced by an improvement operator. Under this view, the standard method—based on the identity transformation—serves as the starting point, which is then mapped to a more preferred transformation through the operator until a fixed point is reached. Formally, we denote $h^1 \succeq h^2$ if $h^1$ is preferred over $h^2$ under the objective defined in Theorem 3.5.

**Corollary 3.6.** *Define the operator $G$ as:*

$$G(h)(s; D, X) = h(w(D, X); D, X)\mathbb{I}(s \geq w(D, X))$$

*Under appropriate conditions, we have $G$ is an improvement and idempotent operator:*

$$G(h) \succeq h \quad and \quad G(G(h)) = G(h) \quad \forall h \in \bar{\mathcal{H}}$$

By applying the operator $G$ to the baseline identity transformation $h(s; D, X) = s$, where convergence is achieved in a single iteration due to idempotence, we derive:

$$G(s)(s; D, X) = w(D, X)\mathbb{I}(s \geq w(D, X)) \succeq s \quad (21)$$

We denote this transformation as $\mathbb{I}_w$. Unlike the optimal solution in Theorem 3.5, $\mathbb{I}_w$ is computationally efficient and requires no prior knowledge of conditional probabilities. Despite its simplicity, $\mathbb{I}_w$ retains the core structural advantages of the theoretical optimum and significantly narrows the coverage gap, as evidenced by our experimental results. Notably, since $\mathbb{I}_w$ is derived through an improvement operator acting directly on the identity mapping, it is theoretically guaranteed to outperform the baseline method. Although monotonically non-decreasing, this transformation can be uniformly approximated by functions in $\mathcal{H}$ to yield arbitrarily close miscoverage levels (see Appendix I).

For the specific case where the transformation is $\mathbb{I}_w$, we summarize the ST-BCP procedure in Algorithm 1. If the prediction set size constraint rule $\mathcal{T}$ depends on the calibration set and the test point feature, then traversal of the label space is required in ST-BCP, which causes the computing time to increase linearly with the size of the label space. Although ST-BCP will bring more computing time costs, it is worthwhile in practical applications. For instance, in automated medical diagnosis, the cost brought about by manual intervention due to excessive conservatism often exceeds the cost of computing time.

---

**Algorithm 1** ST-BCP: $h(s; D, X) = \mathbb{I}_w$

---

1: **Input:** Calibration set $D_{cal}$, test feature $X_{n+1}$, score function $S$ and size constraint rule $\mathcal{T}$,
2: **Output:** Prediction set $C_h^{\tilde{\alpha}_{n+1}(h)}(X_{n+1})$ and estimated coverage bound $\hat{\alpha}^{LOO}(h)$
3: **for** $i = 1$ **to** $n$ **do**
4:     Compute $S_i$, $\mathcal{T}_i$, $w_i$
5:     Compute $h_i = w_i \mathbb{I}(S_i \geq w_i)$
6: **end for**
7: Compute $\mathcal{T}_{n+1}$ and $w_{n+1}$
8: **for** $y \in \mathcal{Y}$ **do**
9:     **for** $i = 1$ **to** $n$ **do**
10:         Compute $\mathcal{T}(D_i^y, X_i)$ and $w(D_i^y, X_i)$
11:         Compute $h_i^y = w(D_i^y, X_i)\mathbb{I}(S_i \geq w(D_i^y, X_i))$
12:     **end for**
13:     Compute $h_{n+1}(y) = w_{n+1}\mathbb{I}(S(X_{n+1}, y) \geq w_{n+1})$
14:     Compute $E^{n+1}(y, h)$ using Eq. (9)
15: **end for**
16: Compute $\tilde{\alpha}_{n+1}(h)$ using Eq. (19)
17: Construct $C_h^{\tilde{\alpha}_{n+1}(h)}(X_{n+1})$ using Eq. (10)
18: **for** $i = 1$ **to** $n$ **do**
19:     Compute $\tilde{\alpha}_i(h)$ using Eq. (20)
20: **end for**
21: Compute $\hat{\alpha}^{LOO}(h)$ using Eq. (15)

---

**Probability Theory Insight.** While our optimization process focuses solely on minimizing the expected miscoverage level $\mathbb{E}[\tilde{\alpha}(h)]$, the resulting optimal transformation $\mathbb{I}_w$ naturally yields a step-function structure. This outcome aligns with a fundamental insight from probability theory: Markov's inequality reaches its tightest state when the random variable is a two-point distribution. Although Markov's inequality was not explicitly considered during derivation, ST-BCP reshapes the score distribution into a two-point structure when $w(D, X)$ is concentrated—a necessary condition for the first-order Taylor approximation (see Appendix E). Further, assuming the average term is concentrated, the e-variable is also transformed into a two-point distribution, aligning with the tightest state of Markov's inequality to yield a tighter coverage bound.

# 4. Experiments

## 4.1. Experimental Setup

**Datasets and Models.** We consider three prominent datasets in our experiments: CIFAR-10, CIFAR-100 (Krizhevsky et al., 2009), and Tiny-ImageNet (Le & Yang, 2015) which are common benchmarks. To evaluate performance across various architectures, we employ several standard image classifiers, including MobileNet (Howard et al., 2019), ResNet (He et al., 2016), EfficientNet (Tan & Le, 2019), and DenseNet (Huang et al., 2017). All classifier models were trained on the full training sets utilizing standard data augmentation techniques, such as random flipping (Yang et al., 2022) and Mixup (Zhang et al., 2017).

**Implementation Details.** We conduct all experiments using the trained models as fixed classifiers. In each experimental round, $n + 1$ samples are randomly drawn without replacement from the test set; these samples are then partitioned into a calibration set of size $n$ and a single test point. To ensure statistical reliability and robustness, we aggregate the reported metrics over $M = 500$ independent random trials. Throughout the experiments, we consistently employ the standard cross-entropy loss, which is also the loss function used for training, as the non-conformity score function. In addition, we have supplemented experiments under different non-conformity score functions in the Appendix.

**Size Constraint Rules.** To rigorously evaluate the adaptability of our method, we implement three distinct types of size constraint rules, denoted as $\mathcal{T}$: constant, feature-dependent, and feature- and calibration-set-dependent. For the non-constant scenarios, we quantify the difficulty of an instance using entropy and map it to the upper bound of the prediction set size. Specifically, the feature-dependent rule derives uncertainty directly from the classifier's softmax output. In contrast, the feature- and calibration-set-dependent rule estimates local uncertainty through a multistage process: first, we project feature representations into a lower-dimensional space using Principal Component Analysis (PCA); subsequently, we compute the entropy of the empirical label distribution formed by the $k$-nearest neighbors within the calibration set. Detailed formulations for both $\mathcal{T}(X)$ and $\mathcal{T}(D, X)$ are provided in Appendix C.

**Evaluation Metrics.** We evaluate the performance of our method by measuring the following metrics. (1) MisCov: the empirical miscoverage; (2) $\mathbb{E}[\tilde{\alpha}_{n+1}(h)]$: the empirical expectation of the miscoverage level; (3) MSE: the Mean Squared Error between the LOO estimator and $\mathbb{E}[\tilde{\alpha}_{n+1}(h)]$; (4) GAP: the coverage gap, defined as the average absolute difference between the LOO estimator and (MisCov); (5) STD: the standard deviation of the LOO estimator across independent trials. The detailed calculation method of the evaluation metrics is shown in Appendix B.

*Table 1.* Performance comparison between BCP($h = s$) and our method ST-BCP($h = \mathbb{I}_w$) under different size constraint rules $\mathcal{T}$. The results are obtained using the ResNet50 model trained on the CIFAR-10 dataset with a calibration set size of $n = 200$. ↓ indicates smaller values are better, and **bold** numbers represent superior results.

| $\mathcal{T}$ | MisCov(%) | STD(%) ↓ | MSE($\times 10^4$) ↓ | GAP(%) ↓ |
|---|---|---|---|---|
| | | | BCP / **ST-BCP (ours)** | |
| 1 | 7.13 / 7.13 | 1.71 / **1.38** | 2.94 / **1.91** | 2.40 / **1.90** |
| 2 | 2.26 / 2.26 | 1.28 / **0.89** | 1.63 / **0.79** | 5.38 / **0.72** |
| 3 | 1.45 / 1.45 | 1.16 / **0.66** | 1.34 / **0.43** | 5.64 / **0.52** |
| $\mathcal{T}(X)$ | 4.27 / 4.27 | 1.43 / **1.13** | 2.04 / **1.28** | 4.04 / **1.17** |
| $\mathcal{T}(D, X)$ | 5.23 / 5.23 | 1.55 / **1.23** | 2.41 / **1.54** | 3.57 / **1.27** |
| Average | 4.07 / 4.07 | 1.43 / **1.06** | 2.07 / **1.19** | 4.21 / **1.12** |

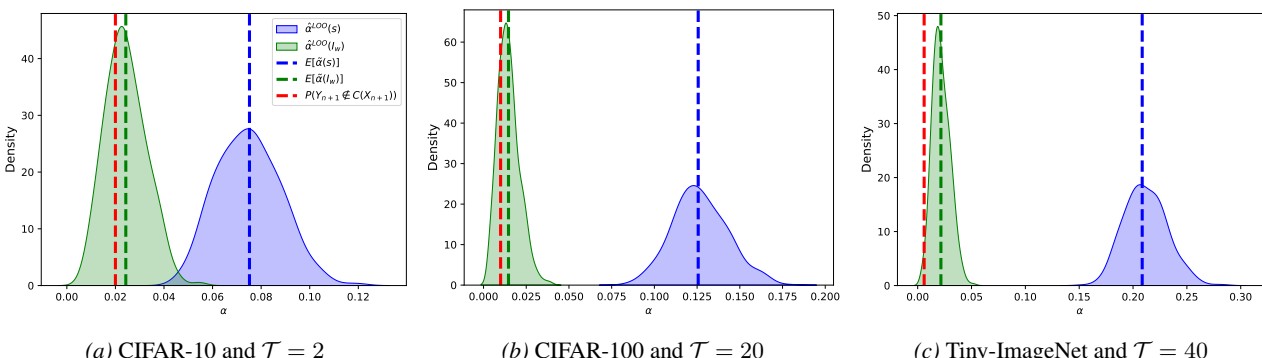

*(a)* CIFAR-10 and $\mathcal{T} = 2$      *(b)* CIFAR-100 and $\mathcal{T} = 20$      *(c)* Tiny-ImageNet and $\mathcal{T} = 40$

*Figure 1.* Performance comparison between BCP ($h = s$) and our method ST-BCP ($h = \mathbb{I}_w$) under different datasets. We present kernel density estimation (KDE) plots of the LOO estimator, the empirical miscoverage, and the empirical expectation miscoverage level. All results are obtained using a ResNet50 model under the specified size constraint rule with a calibration set size $n = 200$. **Note**: Distributions that are more concentrated and closer to the empirical coverage indicate superior performance.

## 4.2. Results

**Reduction of Coverage Gap and Prediction Set Invariance.** Table 1 substantiates the effectiveness of our proposed method. Across various size constraint rules, the gap between the LOO estimator and the empirical miscoverage is significantly reduced, while the asymptotic efficiency (MSE) and estimator stability (STD) actually outperform the baseline. Notably, as highlighted in Remark 3.3, the prediction sets are only potentially subject to change when the size constraint depends on both the features and the calibration set. In this case, the empirical miscoverage remains identical before and after the transformation, even with a relatively small sample size $n$. This observation provides strong empirical support for Theorem 3.2, confirming that the prediction sets remain invariant under the proposed transformation.

**Concentration of the LOO Estimator.** Figure 1 presents the kernel density estimation (KDE) of the LOO estimator, contrasting the identity transformation($h = s$) with the proposed transformation $h = \mathbb{I}_w$ across diverse datasets. Visu-

ally, the distribution under $\mathbb{I}_w$ exhibits a sharp concentration, particularly on the CIFAR-100 and Tiny-ImageNet benchmarks. This concentration signifies the statistical stability of the transformed estimator, which maintains low variance relative to the baseline. Meanwhile, the gap between the LOO estimator and empirical miscoverage is further reduced after the score transformation.

**LOO Estimator Consistency and Sample Efficiency.** While the MSE reported in Table 1 is already minimal, we further evaluate the consistency of the LOO estimator with respect to $\mathbb{E}[\tilde{\alpha}_{n+1}^{pe}(h)]$. Figure 2 illustrates the MSE and STD of the LOO estimator for both the BCP and ST-BCP across varying calibration set sizes. As $n$ increases, both metrics exhibit a clear downward trend, with nearly identical convergence rates and magnitudes for both transformations. Notably, our proposed ST-BCP consistently outperforms the baseline, exhibiting lower MSE and STD across the entire range of $n$. Specifically, when $n = 800$, the MSE of ST-BCP reaches the order of $10^{-5}$, demonstrating superior sample efficiency. This empirical evidence strongly

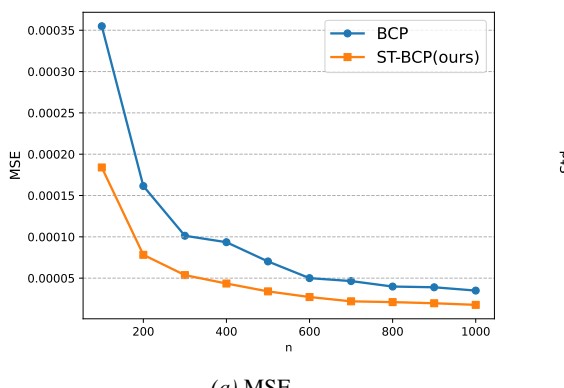 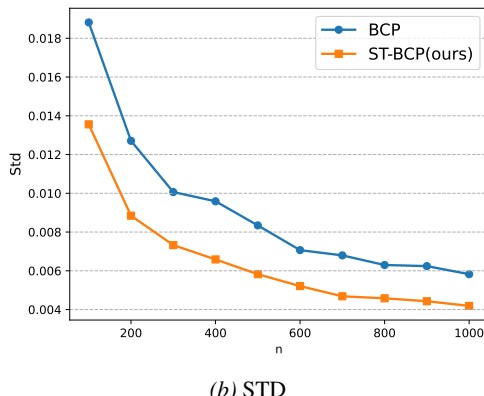

*(a)* MSE      *(b)* STD

*Figure 2.* Comparison of LOO estimator convergence rate and stability between BCP and ST-BCP across different calibration set sizes $n$. We report the MSE and STD that are obtained using a ResNet50 model on the CIFAR-10 dataset under the size constraint rule $\mathcal{T} = 2$. As $n$ increases, our method ST-BCP ($h = \mathbb{I}_w$) consistently outperforms the baseline BCP ($h = s$).

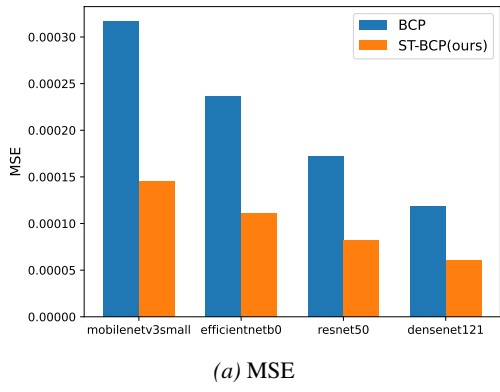 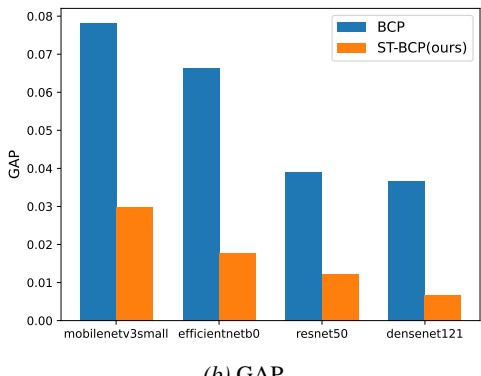

*(a)* MSE      *(b)* GAP

*Figure 3.* Performance comparison between BCP($h = s$) and ST-BCP($h = \mathbb{I}_w$) under different models. We report the MSE and GAP that are obtained on the CIFAR-10 dataset with the calibration set size of $n = 200$ under the size constraint rule $\mathcal{T} = 2$ Among these models, our method ST-BCP($h = \mathbb{I}_w$) has always outperformed the baseline BCP($h = s$).

supports the practical consistency and enhanced stability of the LOO estimator under the proposed transformation.

**ST-BCP Works Well with Different Models.** Figure 3 presents the MSE and GAP across various model architectures. The results demonstrate that ST-BCP consistently and significantly reduces the GAP and MSE compared to the baseline BCP. For instance, with the DenseNet121 model, our method suppresses GAP by approximately 80% (dropping from 4% to 0.7%) and nearly doubles the MSE (from $1.3 \times 10^{-4}$ to $0.6 \times 10^{-4}$). Across all evaluated model architectures, ST-BCP consistently maintains GAD below 3%, whereas the baseline frequently exceeds 4%. This uniform performance gain across diverse models underscores the robustness and broad applicability of our proposed method.

## 5. Related Work

Conformal prediction (Vovk et al., 2005) is a distribution-free uncertainty quantification (Abdar et al., 2021) framework that constructs prediction sets with finite-sample marginal coverage guarantees. It has been widely applied across diverse fields, such as regression (Lei & Wasserman, 2014; Romano et al., 2019), classification (Sadinle et al., 2019), time series (Xu & Xie, 2023), large language models (Kumar et al., 2023; Cherian et al., 2024; Su et al., 2024), and diffusion models (Teneggi et al., 2023). In recent years, numerous methodological advancements have emerged, including enhancing adaptivity (Gauthier et al., 2025a; Szabadváry & Löfström, 2026; Kiyani et al., 2024), designing efficient non-conformity score functions (Huang et al., 2023; Angelopoulos et al., 2020), and modifying the training structure (Liu et al., 2024; Stutz et al., 2021). Research has also focused on achieving conditional coverage guarantees, such as class-conditional coverage for specific subgroups (Jung et al., 2022; Ding et al., 2023) and local coverage for subsets of measurable function classes (Gibbs et al., 2025). Additionally, techniques like weighting (Barber et al., 2023) and online adaptation (Gibbs & Candes, 2021) have been developed to handle cases where the exchangeability assumption is violated.

Conformal e-prediction (Vovk, 2025) employs e-values as the fundamental tool for uncertainty quantification While p-value-based methods are traditionally favored for their statistical efficiency and tighter coverage bounds (Angelopoulos et al., 2024), the e-value framework offers superior mathematical flexibility and structural advantages that extend the scope of CP (Gauthier et al., 2025c). Specifically, e-values enable novel application scenarios typically inaccessible to p-values. It provides guaranteed validity for cross-conformal predictors, a property that standard methods satisfy only empirically. A key structural property is that the arithmetic mean of e-values remains a valid e-value (Vovk & Wang, 2021), which facilitates de-randomization and robust model aggregation. For instance, this property has been leveraged to develop symmetric aggregation strategies (Alami et al., 2025), which combine non-conformity scores from multiple predictors into efficient uncertainty sets. Furthermore, e-values support post-hoc validity (Koning, 2024) and allow for the construction of conformal sets with data-dependent miscoverage levels (Grünwald, 2024; Csillag et al., 2025).

## 6. Conclusion

In this paper, we propose a novel method by introducing data-dependent transformation of non-conformity scores. Our approach narrows the gap between empirical coverage and estimated coverage bound while strictly preserving data exchangeability and the consistency of the LOO estimator. We derive the theoretically optimal transformation under monotonicity constraints and provide a computable approximation that is preferred over the baseline identity transformation (original BCP). Experimental results across various datasets, models, and size constraint rules demonstrate that our method significantly narrows the coverage gap, maintaining convergence and stability comparable to the original BCP. This work offers an efficient solution for uncertainty quantification in scenarios requiring controlled prediction set sizes.

**Limitations.** The theoretical assumptions in our method are complex, particularly when size constraints depend on both the feature and the calibration set. Furthermore, while ensuring estimator consistency, our derivation does not optimize for estimation efficiency. Future work can focus on relaxing these assumptions or proving that the transformed estimator achieves lower variance than the baseline.

## Acknowledgement

This research is supported by Guangdong Basic and Applied Basic Research Foundation (Grant No. 2026A1515011367) and the SUSTech-NUS Joint Research Program. This project is also supported by the Jiangsu Provincial Key Discipline Construction Project (Statistics) and open project of Joint Lab for Statistics and Finance (Grant No. 2025JLSF101). We gratefully acknowledge the support of the Center for Computational Science and Engineering at the Southern University of Science and Technology for our research.

## Impact Statement

This paper presents work whose goal is to advance the field of machine learning. There are many potential societal consequences of our work, none of which we feel must be specifically highlighted here.

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

*Table 2.* Performance comparison between BCP($h = s$) and our method ST-BCP($h = \mathbb{I}_w$) under different non-conformity score function $S(X, Y)$. The results are obtained using the ResNet50 model trained on the CIFAR-10 dataset with a calibration set size of $n = 200$ under the size constraint rule $\mathcal{T} = 2$. ↓ indicates smaller values are better, and **bold** numbers represent superior results.

| | BCP **/ ST-BCP (ours)** | | | |
| --- | --- | --- | --- | --- |
| Score Function | MisCov(%) | STD(%) ↓ | MSE($\times 10^4$) ↓ | GAP(%) ↓ |
| APS | 2.80 / 2.80 | 2.02 / **1.07** | 4.07 / **1.14** | 48.2 / **0.84** |
| Cross-Entropy | 2.26 / 2.26 | 1.28 / **0.89** | 1.63 / **0.79** | 5.38 / **0.72** |
| Rank | 2.20 / 2.20 | 1.34 / **1.12** | 1.79 / **1.26** | 35.6 / **1.14** |
| THR | 2.58 / 2.58 | 1.64 / **1.09** | 2.69 / **1.19** | 13.7 / **0.92** |

## A. The Improvement Effects of Different Non-Conformity Score Functions

We conducted various experiments with the non-conformity score function to demonstrate that the validity of ST-BCP is insensitive to the selection of it. In table 1, we observe that ST-BCP consistently reduces the coverage gap (GAP) and enhances the stability of the LOO estimator. The improvement is particularly substantial when the baseline score performs poorly. For example, ST-BCP drastically reduces the GAP of the APS from 48.2% to 0.84%.

## B. Evaluation Metrics

To quantify the performance of ST-BCP across $M$ independent experimental trials, we employ a suite of statistical metrics. In the $m$-th trial, let $(X_{n+1}^{(m)}, Y_{n+1}^{(m)})$ denote the test point, $\hat{\alpha}_{(m)}^{LOO}(h)$ the leave-one-out estimator, and $C(X_{n+1}^{(m)}; h)$ the resulting prediction set. The following metrics are computed to evaluate estimation precision and coverage stability:

Empirical Miscoverage (MisCov): The observed frequency with which the true label falls outside the prediction set.

$$\text{MisCov} = \frac{1}{M} \sum_{m=1}^{M} \mathbb{I}(Y_{n+1}^{(m)} \notin C(X_{n+1}^{(m)}; h))$$

Empirical Expectation of Miscoverage Level: The average data-dependent miscoverage level across all trials.

$$\mathbb{E}[\tilde{\alpha}_{n+1}(h)] = \frac{1}{M} \sum_{m=1}^{M} \tilde{\alpha}_{n+1}^{(m)}(h)$$

Mean Squared Error (MSE): Quantify the rate at which the LOO estimator converges to $\mathbb{E}[\tilde{\alpha}_{n+1}(h)]$

$$\text{MSE} = \frac{1}{M} \sum_{m=1}^{M} \left( \hat{\alpha}_{(m)}^{LOO}(h) - \mathbb{E}[\tilde{\alpha}_{n+1}(h)] \right)^2$$

Coverage Gap (GAP): The average absolute gap between the estimated bound and empirical miscoverage.

$$\text{GAP} = \frac{1}{M} \sum_{m=1}^{M} \left| \hat{\alpha}_{(m)}^{LOO}(h) - \text{MisCov} \right|$$

Sample standard deviation (STD): The stability of the LOO estimator.

$$\text{STD} = std(\hat{\alpha}_{(1)}^{LOO}(h), \ldots, \hat{\alpha}_{(M)}^{LOO}(h))$$

## C. Adaptive Size Constraint Rule $\mathcal{T}$

### C.1. Entropy-Based Construction

When $\mathcal{T}$ is not a constant, it is typically desirable for difficult data points to receive large prediction sets. Since entropy is a standard measure for quantifying uncertainty, we employ it to adjust $\mathcal{T}$; similarly, the Gini index could also be applied. We

---

**Algorithm 2** single size constraint rule $\mathcal{T}(D, X)$

---

1: **Input:** Dataset $D = \{(X_i, Y_i)\}_{i=1}^n$, feature $X = X_{n+1}$ , parameters $\mathcal{T}_{min}, \mathcal{T}_{max}, k, p$
2: **Output:** $\mathcal{T}(D, X_{n+1})$
3: **for** $j = 1$ **to** $n + 1$ **do**
4:     Compute the $k$-nearest neighbors of $X_j$ among $\{X_i\}_{i=1}^{n+1}$, denote the index set by $N_j^k$
5:     Compute the empirical distribution $\pi_j^k$ of $\{Y_i\}_{i \in N_j^k}$,
6:     Compute the entropy $EN_j = -\sum_{y \in \mathcal{Y}} \pi_j^k(y) \log \pi_j^k(y)$
7: **end for**
8: Compute $EN_{max} = max(EN_1, ..., EN_{n+1}); EN_{min} = min(EN_1, ..., EN_{n+1})$
9: **for** $l = 1$ **to** $\mathcal{T}_{max} - \mathcal{T}_{min} + 1$ **do**
10:     Compute bin thresholds $bins(l) = EN_{min} + (EN_{max} - EN_{min}) \left( \frac{l-1}{L-1} \right)^p$
11: **end for**
12: Compute $\mathcal{T}(D, X) = b(EN_{n+1}) = \mathcal{T}_{min} - 1 + \sum_{l=1}^{L} \mathbb{I}(EN_{n+1} \geq bins(l))$

---

focus on classification tasks where $\mathcal{T}$ is discrete. To this end, a binning function $b$ is used to map the entropy value to a bounded positive integer. One may also impose $0 < \mathcal{T}_{min} < \mathcal{T} < \mathcal{T}_{max} \leq |\mathcal{Y}|$ to further constraints on the output. Let $EN \in \mathbb{R}^+$ be the entropy; then:

$$\mathcal{T}(D, X) = b(EN), \quad b : \mathbb{R}^+ \to \{\mathcal{T}_{min}, ..., \mathcal{T}_{max}\}$$

We consider the case where $\mathcal{T}$ depends on both the calibration set $D = D_{cal} = \{(X_i, Y_i)\}_{i=1}^n$ and the test point feature $X = X_{n+1}$. Taking $\mathcal{T}(D_{cal}, X_{n+1})$ as an example, the binning function $b$ can be constructed based on the relative magnitude of the test point's entropy $EN_{n+1} = EN(D_{cal}; X_{n+1})$ within the set of pseudo-calibration entropy $\{EN_i = EN(D_{cal}, X_i)\}_{i=1}^n$. In the experiment corresponding to Table 1 (where $\mathcal{T} = \mathcal{T}(D, X)$), the maximum entropy $EN_{max}$ and minimum entropy $EN_{min}$ are derived from the combined set of entropy $\{EN_1, \ldots, EN_n, EN_{n+1}\}$ to serve as the boundaries for binning.

To calculate entropy, we employ a $k$-nearest neighbor (k-NN) approach following PCA-based dimensionality reduction. A pre-trained ResNet-18 model is utilized for feature extraction; specifically, by removing the final fully connected layer, we extract 512-dimensional feature representations from the global average pooling layer. These high-dimensional features for both the calibration set and the test point are subsequently projected into a two-dimensional space via Principal Component Analysis (PCA). Following this, we compute the label distribution $\pi^k(D, X)$ based on the $k$ nearest neighbors of the reduced feature $X$ within the reduced calibration set $D$. The complete procedure for deriving $\mathcal{T}(D, X)$ where $D = \{(X_i, Y_i)\}_{i=1}^n$ and $X = X_{n+1}$ representing the processed feature representations is detailed in Algorithm 2.

Since the score transformation is dependent on $\mathcal{T}$, multiple size constraints must be evaluated. Although their inputs vary in form, Algorithm 3 provides a unified approach for organizing these diverse inputs. Specifically, we let $Q(D, X)$ denote the feature extraction and dimensionality reduction process applied to a dataset–query pair; the corresponding $\mathcal{T}$ is then computed by invoking Algorithm 2.

When $\mathcal{T}$ is allowed to depend only on the test point feature X, the entropy is computed directly from the model's softmax output $\pi(X)$. Specifically, we define

$$EN_{max} = log(|\mathcal{Y}|), \quad EN_{min} = 0, \quad EN(X) = -\sum_{y \in \mathcal{Y}} \pi_y(X) log(\pi_y(X))$$

Since this setting does not involve any dataset-level input $D$, the computation is a simplified special case of $\mathcal{T}(D, X)$. Consequently, both the computational procedure and the number of required size constraints are substantially reduced, and we omit the explicit algorithmic description for brevity.

### C.2. Sensitivity Analysis of the k-NN Parameter $k$

The adaptive size constraint rule $\mathcal{T}(D, X)$ relies on the label distribution estimated from the $k$-nearest neighbors in the reduced feature space. Therefore, it is natural to examine the sensitivity of $\mathcal{T}(D, X)$ and the resulting transformed threshold $w(D, X)$ with respect to the choice of $k$.

---

**Algorithm 3** multiple size constraint rule $\mathcal{T}(Q(D, X))$

---

    **Input:** Dataset $D_{cal} = \{(X_i, Y_i)\}_{i=1}^n$, test feature $X_{n+1}$, parameters $\mathcal{T}_{min}, \mathcal{T}_{max}, k, p$

2:  **Output:** $\mathcal{T}(Q(D_i, X_i)), i = 1, \cdots, n + 1$ and $\mathcal{T}(Q(D_i^y, X_i)), i = 1, \cdots, n; y \in \mathcal{Y}$

    **for** $i = 1$ **to** $n + 1$ **do**

4:    Extract a 512-dimensional feature vector $X_i^{ex}$ from $X_i$ using pre-trained ResNet-18

    **end for**

6:  Apply PCA to $\{X_i^{ex}\}_{i=1}^{n+1}$ and obtain $\{X_i^{p1}\}_{i=1}^{n+1}, X_i^{p1} \in \mathbb{R}^2$

    Apply PCA to $\{X_i^{ex}\}_{i=1}^{n}$ and obtain $\{X_i^{p2}\}_{i=1}^{n}, X_i^{p2} \in \mathbb{R}^2$

8:  **for** $y \in \mathcal{Y}$ **do**

    **for** $i = 1$ **to** $n$ **do**

10:      Compute $\mathcal{T}(Q(D_i^y, X_i))$, where $Q(D_i^y, X_i) = (\{(X_{n+1}^{p1}, y)\} \cup \{(X_j^{p1}, Y_j)\}_{j=1, j\neq i}^{n}, X_j^{p1})$

    **end for**

12: **end for**

    Compute $\mathcal{T}(Q(D_{n+1}, X_{n+1}))$, where $Q(D_{n+1}, X_{n+1}) = (\{(X_i^{p1}, Y_i)\}_{i=1}^{n}, X_{n+1}^{p1})$

14: **for** $i = 1$ **to** $n$ **do**

    Compute $\mathcal{T}(Q(D_i, X_i))$, where $Q(D_i, X_i) = (\{(X_j^{p2}, Y_j)\}_{j=1, j\neq i}^{n}, X_i^{p2})$

16: **end for**

---

In the corresponding experiment reported in Table 1, we set $k = 20$. To evaluate robustness, we additionally considered $k \in \{20, 21, 30, 70\}$, while fixing the calibration set and the test point. Table 3 reports the empirical distribution of the adaptive size constraint $\mathcal{T}(D, X)$ under different values of $k$.

*Table 3.* Distribution of adaptive size constraint $\mathcal{T}(D, X)$ under different values of $k$.

|  | $\mathcal{T} = 1$ | $\mathcal{T} = 2$ | $\mathcal{T} = 3$ |
|---|---|---|---|
| $k = 20$ | 0.68 | 0.31 | 0.01 |
| $k = 21$ | 0.63 | 0.36 | 0.01 |
| $k = 30$ | 0.55 | 0.45 | 0 |
| $k = 70$ | 0.67 | 0.33 | 0 |

To further quantify the sensitivity of $\mathcal{T}(D, X)$ and $w(D, X)$, we computed the absolute variation relative to the baseline choice $k = 20$. Specifically, letting $\Delta\mathcal{T} = \mathcal{T}_k(D, X) - \mathcal{T}_{20}(D, X), \quad \Delta w = w_k(D, X) - w_{20}(D, X)$. We report their empirical distributions over calibration and test points in Table 4 and Table 5.

*Table 4.* Distribution of absolute changes in $\mathcal{T}(D, X)$ relative to $k = 20$.

|  | $|\Delta\mathcal{T}| = 0$ | $|\Delta\mathcal{T}| = 1$ | $|\Delta\mathcal{T}| = 2$ |
|---|---|---|---|
| $k = 21$ | 0.92 | 0.085 | 0 |
| $k = 30$ | 0.74 | 0.26 | 0 |
| $k = 70$ | 0.67 | 0.32 | 0.005 |

*Table 5.* Distribution of absolute changes in $w(D, X)$ relative to $k = 20$.

|  | $|\Delta w| = 0$ | $0 < |\Delta w| \leq 0.5$ | $0.5 < |\Delta w| \leq 1$ | $1 < |\Delta w|$ |
|---|---|---|---|---|
| $k = 21$ | 0.92 | 0.069 | 0.005 | 0.009 |
| $k = 30$ | 0.74 | 0.18 | 0.020 | 0.059 |
| $k = 70$ | 0.67 | 0.21 | 0.045 | 0.075 |

The results indicate that $\mathcal{T}(D, X)$ is relatively stable with respect to the choice of $k$. Even when $k$ increases from 20 to 70, most adaptive size constraints remain unchanged. Moreover, when changes occur, they are typically of magnitude one. Since the transformed threshold $w(D, X)$ depends on $\mathcal{T}(D, X)$, this stability propagates to the score transformation. In particular, the deviation of $w(D, X)$ is predominantly zero or remains very small across different choices of $k$.

## D. The Details of $w(D, X)$

In some cases, $w(D, X)$ can be obtained in closed form for faster computation, as opposed to using a binary search. For classification tasks, denote $X$'s score sequence as $S(X, y)_{y \in \mathcal{Y}}$. Let:

$$S(X, y)_{(1)} \leq \cdots \leq S(X, y)_{(|\mathcal{Y}|)} < S(X, y)_{(|\mathcal{Y}|+1)} \overset{\text{def}}{=} +\infty$$

Then:

$$w(D, X) = \sup\{l > 0 : |\{y : S(X, y) < l\}| \leq \mathcal{T}(D, X)\} = S(X, y)_{(\mathcal{T}(D, X)+1)}$$

For regression tasks, assume that the score function is $S(X, Y) = ||f(X) - Y||_q$ and $\mathcal{Y} = \mathbb{R}^d$, $q > 0$, $l > 0$ then,

$$|\{y : S(X, y) < l\}| = |B_q^d(l)|; B_q^d(l) = \{y \in \mathbb{R}^d : ||y||_p < l\}$$

$$|B_q^d(l)| = \frac{\left[2\,\Gamma(1 + 1/q)\right]^d}{\Gamma(d/q + 1)} l^d$$

$B_q^d(l)$ is the ball of radius $l$ in the normed linear space $(\mathbb{R}^d, ||\cdot||_q)$. Computing its volume is a classical result. We briefly outline the derivation (Wang, 2005): using symmetry and variable substitution, the integration domain is transformed into a simplex, after which the Dirichlet integral formula is applied to obtain the volume.

$$|B_q^d(l)| \leq \mathcal{T}(D, X) \Leftrightarrow l \leq \frac{\left[\mathcal{T}(D, X)\Gamma(d/q + 1)\right]^{1/d}}{2\Gamma(1 + 1/q)}$$

$$w(D, X) = \sup\{l > 0 : |B_q^d(l)| \leq \mathcal{T}(D, X)\} = \frac{\left[\mathcal{T}(D, X)\Gamma(d/q + 1)\right]^{1/d}}{2\Gamma(1 + 1/q)}$$

## E. Discussion of Taylor Approximation

A fundamental limitation of BCP is the coverage guarantee. Specifically, the BCP guarantee relies on a first-order Taylor approximation to relate the data-dependent miscoverage level $\tilde{\alpha}_{n+1}$ to the actual coverage.

$$1 \underset{\text{Markov Error}}{\geq} \mathbb{E}\left[\frac{\mathbb{P}\left(Y_{n+1} \notin C_h^{\tilde{\alpha}_{n+1}(h)}(X_{n+1}) \mid \tilde{\alpha}_{n+1}(h)\right)}{\tilde{\alpha}_{n+1}(h)}\right] \underset{\text{Taylor Error}}{\approx} \frac{\mathbb{E}\left[\mathbb{P}\left(Y_{n+1} \notin C_h^{\tilde{\alpha}_{n+1}(h)}(X_{n+1}) \mid \tilde{\alpha}_{n+1}(h)\right)\right]}{\mathbb{E}[\tilde{\alpha}_{n+1}(h)]}$$

$$= \frac{\mathbb{P}\left(Y_{n+1} \notin C_h^{\tilde{\alpha}_{n+1}(h)}(X_{n+1})\right)}{\mathbb{E}[\tilde{\alpha}_{n+1}(h)]} \Rightarrow \mathbb{P}\left(Y_{n+1} \in C_h^{\tilde{\alpha}_{n+1}(h)}(X_{n+1})\right) \geq 1 - \mathbb{E}[\tilde{\alpha}_{n+1}(h)]$$

In this section, we elaborate on these issues in detail, including the necessity of Taylor approximations, whether the proposed transformations violate the original coverage guarantee, the specific conditions under which the guarantee may fail, and potential remedies. For the following derivation, let $X$ and $Y$ be non-negative random variables. Utilizing the Taylor series expansion of an infinite series, we obtain:

$$\mathbb{E}\left[\frac{X}{Y}\right] = \frac{\mathbb{E}[X]}{\mathbb{E}[Y]}\left[\sum_{k=0}^{\infty}(-1)^k \frac{\mathbb{E}[(Y - \mathbb{E}[Y])^k]}{(\mathbb{E}[Y])^k}\right] + \frac{1}{\mathbb{E}[Y]}\left[\sum_{k=0}^{\infty}(-1)^k \frac{\mathbb{E}[(X - \mathbb{E}[X])(Y - \mathbb{E}[Y])^k]}{(\mathbb{E}[Y])^k}\right]$$

The expansion is valid under the following condition.

$$\left|\frac{Y - \mathbb{E}[Y]}{\mathbb{E}[Y]}\right| < 1$$

Therefore, the first-order Taylor approximation

$$\mathbb{E}\left[\frac{X}{Y}\right] \approx \frac{\mathbb{E}[X]}{\mathbb{E}[Y]}$$

is exact if and only if:

$$\frac{1}{\mathbb{E}[Y]} \left[ \sum_{k=1}^{\infty} (-1)^k \frac{\mathbb{E}[X(Y - \mathbb{E}[Y])^k]}{(\mathbb{E}[Y])^k} \right] = 0$$

To simplify the analysis, we focus on the concentration-dominated regime. When $Y$ is highly concentrated and has a strict positive lower bound, the higher-order terms become small, resulting in a small Taylor approximation error. We note that this is only a sufficient condition. Even when $Y$ is not concentrated, the higher-order terms may still cancel due to the dependence structure between $X$ and $Y$. For example, when $X = Y$, the infinite series above is still equal to zero.

In BCP and ST-BCP: $X$ and $Y$ correspond to:

$$X = \mathbb{P}\left(Y_{n+1} \notin C_h^{\tilde{\alpha}_{n+1}(h)}(X_{n+1}) \mid \tilde{\alpha}_{n+1}(h)\right), \quad Y = \tilde{\alpha}_{n+1}(h).$$

And

$$\tilde{\alpha}_{n+1}(h) = \frac{1}{n+1} \left( \frac{\sum_{i=1}^{n} h_i}{h(w_{n+1}; D_{n+1}, X_{n+1})} + 1 \right) = \frac{(\sum_{i=1}^{n} h_i)/(n+1)}{h(w_{n+1}; D_{n+1}, X_{n+1})} + \frac{1}{n+1}$$

Since $h \geq 0$,

$$\tilde{\alpha}_{n+1}(h) \geq \frac{1}{n+1}$$

thus $\tilde{\alpha}_{n+1}(h)$ has a strict positive lower bound $1/(n+1)$. The numerator is an empirical average and is typically concentrated. Therefore, the main source of fluctuation comes from the denominator term:

$$h(w_{n+1}; D_{n+1}, X_{n+1})$$

For both the identity transformation and the proposed $\mathbb{I}_w$ transformation,

$$h(w_{n+1}; D_{n+1}, X_{n+1}) = w_{n+1}.$$

Hence, the proposed transformation does not substantially change the concentration behavior of $\tilde{\alpha}_{n+1}(h)$, and therefore does not substantially change the Taylor approximation error induced by concentration.

If we only consider whether the coverage guarantee holds, Jensen's inequality gives a weaker sufficient condition that does not directly rely on the Taylor approximation:

$$1 \geq \mathbb{E}\left[\frac{\mathbb{P}\left(Y_{n+1} \notin C_h^{\tilde{\alpha}_{n+1}(h)}(X_{n+1}) \mid \tilde{\alpha}_{n+1}(h)\right)}{\tilde{\alpha}_{n+1}(h)}\right]$$

$$\geq \frac{\mathbb{P}\left(Y_{n+1} \notin C_h^{\tilde{\alpha}_{n+1}(h)}(X_{n+1})\right)}{\mathbb{E}[\tilde{\alpha}_{n+1}(h)]} + Cov\left(\mathbb{P}\left(Y_{n+1} \notin C_h^{\tilde{\alpha}_{n+1}(h)}(X_{n+1}) \mid \tilde{\alpha}_{n+1}(h)\right), \frac{1}{\tilde{\alpha}_{n+1}(h)}\right) \Rightarrow$$

$$\mathbb{P}\left(Y_{n+1} \in C_h^{\tilde{\alpha}_{n+1}(h)}(X_{n+1})\right) \geq 1 - \mathbb{E}[\tilde{\alpha}_{n+1}(h)]\left(1 - Cov\left(\mathbb{P}\left(Y_{n+1} \notin C_h^{\tilde{\alpha}_{n+1}(h)}(X_{n+1}) \mid \tilde{\alpha}_{n+1}(h)\right), \frac{1}{\tilde{\alpha}_{n+1}(h)}\right)\right)$$

Therefore,

$$\mathbb{P}\left(Y_{n+1} \in C_h^{\tilde{\alpha}_{n+1}(h)}(X_{n+1})\right) \geq 1 - \mathbb{E}[\tilde{\alpha}_{n+1}(h)]\left(1 - Cov(\cdot)\right)$$

The original coverage guarantee is recovered when the covariance term is approximately zero. This occurs when either the conditional miscoverage probability or $1/\tilde{\alpha}_{n+1}(h)$ is highly concentrated.

The Taylor approximation or the covariance-based guarantee may fail when $\tilde{\alpha}_{n+1}(h)$ is not concentrated. This may occur when the size constraint is strict or when the prediction model is unstable. To address this issue, we propose two potential solutions: (1) implementing a robust transformation to optimize the non-concentrated term $h(w_{n+1}; D_{n+1}, X_{n+1})$; and (2) adopting a method that bypasses the need for Taylor approximations entirely, such as the corrected coverage level approach.

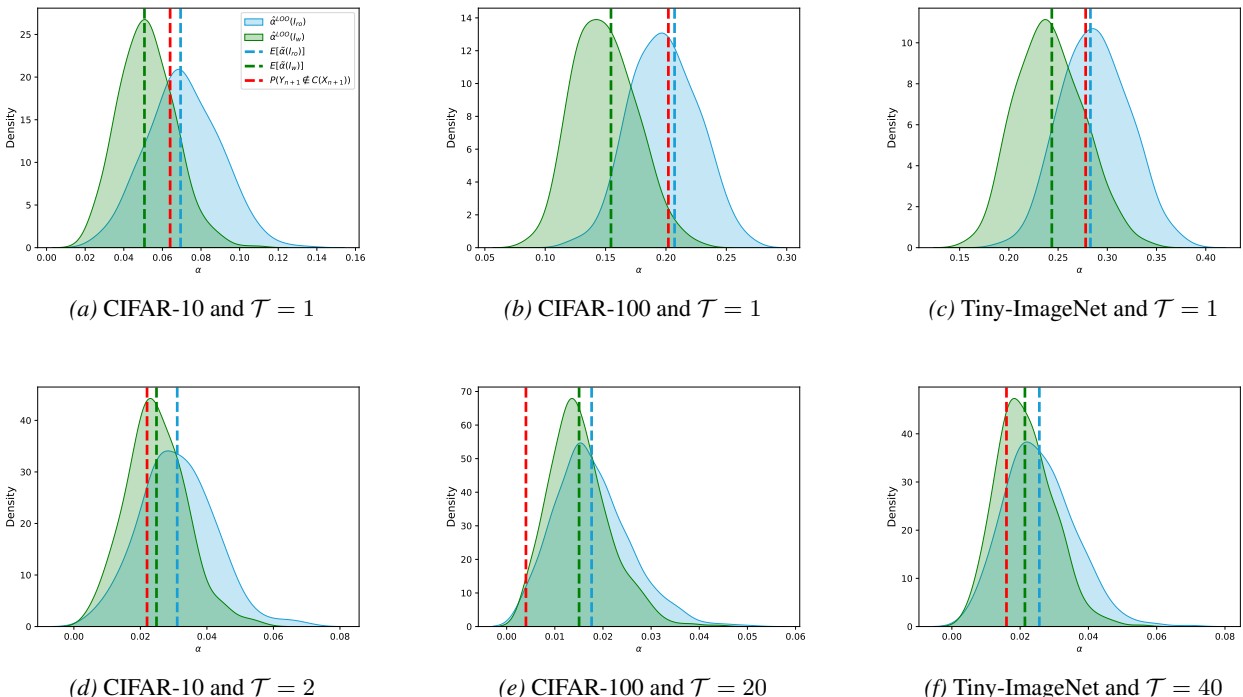

*Figure 4.* Under different datasets, the MisCov, kernel density estimation plots of the empirical distribution of the LOO estimator, and the empirical miscoverage levels of both are presented. The results were obtained with the size constraint rule, using the ResNet50 model and the calibration set size $n = 200$.

To address the lack of concentration in $\tilde{\alpha}_{n+1}(h)$, we can enforce $h(w_{n+1}; D_{n+1}, X_{n+1}) = 1$. This is achieved by removing the $w(D, X)$ coefficient from the previously defined transformation $\mathbb{I}_w = w(D, X)\mathbb{I}(s \geq w(D, X))$. The resulting robust transformation is defined as $h(s; D, X) = \mathbb{I}_{ro} \stackrel{\text{def}}{=} \mathbb{I}(s \geq w(D, X))$.

For the coverage guarantee that does not rely on Taylor approximation(Gauthier et al., 2025b). Here, we provide a brief method, call it the corrected coverage level.

$$\mathbb{P}(Y_{n+1} \notin C_h^{\tilde{\alpha}_{n+1}(h)}(X_{n+1})) = \mathbb{P}(E^{n+1}(Y_{n+1}, h) \geq \frac{1}{\tilde{\alpha}_{n+1}(h)}) \leq \mathbb{E}\left[\tilde{\alpha}_{n+1}(h)E^{n+1}(Y_{n+1}, h)\right]$$

This is similar to conformal e-prediction methods, providing a rigorous coverage guarantee derived from the Markov inequality, which can be obtained without requiring exchangeability. As introduced in the main text, under certain assumptions, this coverage guarantee bound can be directly estimated using the leave-one-out estimator. $\tilde{\alpha}_i(h)E^i(Y_i, h)$ is denoted as $\tilde{b}_i(h)$, $i = 1, \cdots, n + 1$. The corrected leave-one-out estimator $\hat{b}^{LOO}(h)$ is defined as follows:

$$\hat{b}^{LOO}(h) = \frac{1}{n} \sum_{i=1}^{n} \tilde{b}_i(h)$$

In Figure 5 and Figure 4, we compare the corrected coverage level method ($\tilde{b}(\mathbb{I}_w)$) and the robust transformation ($\mathbb{I}_{ro}$) with the original transformation ($\mathbb{I}_w$) under different size constraint regimes. When the size constraint $\mathcal{T}$ is small, both the corrected coverage level method and the robust transformation reduce cases in which the true coverage exceeds the coverage guarantee bound, thereby alleviating the overconfidence of the leave-one-out estimator.

When the size constraint $\mathcal{T}$ is large, the Taylor approximation becomes accurate, and the original transformation is sufficient to maintain the validity of the coverage guarantee. In this case, the corrected coverage level method and the robust transformation become slightly conservative. Moreover, in both regimes, the estimation effect of the leave-one-out estimators produced by the original transformation is consistently better than those produced by the corrected coverage level method and the robust transformation.

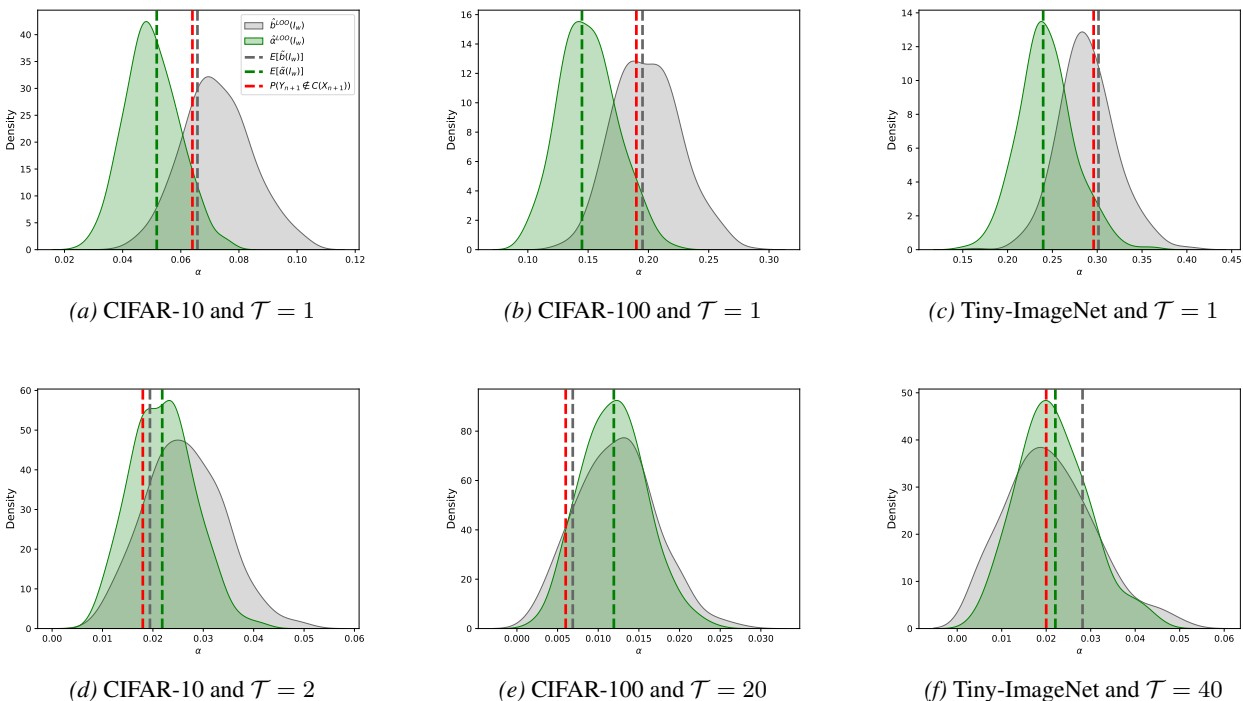

*(a)* CIFAR-10 and $\mathcal{T} = 1$      *(b)* CIFAR-100 and $\mathcal{T} = 1$     *(c)* Tiny-ImageNet and $\mathcal{T} = 1$

*(d)* CIFAR-10 and $\mathcal{T} = 2$     *(e)* CIFAR-100 and $\mathcal{T} = 20$     *(f)* Tiny-ImageNet and $\mathcal{T} = 40$

*Figure 5.* Under different datasets, the MisCov, kernel density estimation plots of the empirical distribution of the LOO estimator, and the empirical miscoverage levels of both are presented. The results were obtained with the size constraint rule, using the ResNet50 model and the calibration set size $n = 200$.

Based on these observations, we recommend using the corrected coverage level method or the robust transformation when the size constraint $\mathcal{T}$ is small and using the original transformation when $\mathcal{T}$ is large.

## F. Proof of Theorem 3.1

**Theorem 3.1.** *Assume that the transformation $h(s; D, X)$ is bounded such that $h(s; D, X) \in [h_{min}, h_{max}]$ with $0 < h_{min} \le h_{max}$, and the calibration set size satisfies $n > h_{max}/h_{min}$. Furthermore, assume that $h$ and the size constraint rule satisfies the following properties almost surely:*

*(P1) For all $y \in \mathcal{Y}$, the sequences $\{h_j\}_{j=1}^n$, $\{h_i^y\}_{i=1}^n$, and $\{h_j(y)\}_{j=1}^n$ all satisfy Chernoff–Hoeffding bounds.*

*(P2) For all $y \in \mathcal{Y}$ have:*

$$|\mathbb{E}[h_i] - \mathbb{E}[h_i^y]| \le \frac{h_{max}}{\sqrt{n}}$$

$$|\mathbb{E}[h_i(y)] - \mathbb{E}[h_{n+1}(y)]| \le \frac{h_{max}}{\sqrt{n}}$$

*(P3) There exists a constant $L > 0$ such that:*

$$\left| \sum_{i=1}^n (\tilde{\alpha}_i(h) - \mathbb{E}[\tilde{\alpha}_{n+1}(h)]) \right| \le L \max_{y \in \mathcal{Y}} \left| \sum_{i=1}^n (E^i(y, h) - E^{n+1}(y, h)) \right|$$

*Then, the leave-one-out estimator satisfies the following:*

$$|\hat{\alpha}^{LOO}(h) - \mathbb{E}[\tilde{\alpha}_{n+1}(h)]| = O_p\left(\frac{1}{\sqrt{n}}\right) \qquad (22)$$

The proof process of this theorem mostly refers to paper (Gauthier et al., 2025b). If you are interested in the proof of this theorem, we strongly recommend that you read the corresponding part first, although it is not necessary.

The Chernoff-Hoeffding Bounds in Property (P1) mean:

$$\mathbb{P}\left(\left|\frac{1}{n}\sum_{i=1}^{n} h_i - \mathbb{E}[h_i]\right| \geq t\right) \leq 2exp\left[\frac{-2nt^2}{(h_{max} - h_{min})^2}\right]$$

$$\mathbb{P}\left(\left|\frac{1}{n}\sum_{i=1}^{n} h_i^y - \mathbb{E}[h_i^y]\right| \geq t\right) \leq 2exp\left[\frac{-2nt^2}{(h_{max} - h_{min})^2}\right], \forall y \in \mathcal{Y}$$

$$\mathbb{P}\left(\left|\frac{1}{n}\sum_{i=1}^{n} h_i(y) - \mathbb{E}[h_i(y)]\right| \geq t\right) \leq 2exp\left[\frac{-2nt^2}{(h_{max} - h_{min})^2}\right], \forall y \in \mathcal{Y}$$

Clearly, $h_1, ..., h_n$ are exchangeable and hence identically distributed. The same holds for $h_1^y, ..., h_n^y$, for all $y \in \mathcal{Y}$.

So we can briefly denote:

$$\mu(h) = \mathbb{E}[h_i], \mu^y(h) = \mathbb{E}[h_i^y], \tilde{E}^i(y, h) = \frac{h^i(y)}{\mu}, i = 1, \cdots, n$$

$$\tilde{E}^{n+1}(y, h) = \frac{h^{n+1}(y)}{\mu^y}; \mathbf{E}^i(h) = (E^i(y, h))_{y \in \mathcal{Y}}, i = 1, \cdots, n+1$$

In the proof, the norm is defined as: $||\mathbf{E}|| = \max_{y \in \mathcal{Y}} |E(y)|$. This definition satisfies the three axioms of the norm.

*Proof.* By Property(P3) and the triangle inequality, we have:

$$|\hat{\alpha}^{LOO}(h) - \mathbb{E}[\tilde{\alpha}_{n+1}(h)]| = \frac{1}{n}\left|\sum_{i=1}^{n}(\tilde{\alpha}_i(h) - \mathbb{E}[\tilde{\alpha}_{n+1}(h)])\right| \leq \frac{L}{n}\left\|\sum_{i=1}^{n}(\mathbf{E}^i(h) - \mathbb{E}[\mathbf{E}^{n+1}(h)])\right\|$$

$$= \frac{L}{n}\left\|\sum_{i=1}^{n}\left[(\mathbf{E}^i(h) - \tilde{\mathbf{E}}^i(h)) + (\tilde{\mathbf{E}}^i(h) - \mathbb{E}[\tilde{\mathbf{E}}^{n+1}(h)]) + (\mathbb{E}[\tilde{\mathbf{E}}^{n+1}(h)] - \mathbb{E}[\mathbf{E}^{n+1}(h)])\right]\right\|$$

$$\leq \frac{L}{n}\left\|\sum_{i=1}^{n}(\mathbf{E}^i(h) - \tilde{\mathbf{E}}^i(h))\right\| + \frac{L}{n}\left\|\sum_{i=1}^{n}(\tilde{\mathbf{E}}^i(h) - \mathbb{E}[\tilde{\mathbf{E}}^{n+1}(h)])\right\|$$

$$+ \frac{L}{n}\left\|\sum_{i=1}^{n}(\mathbb{E}[\tilde{\mathbf{E}}^{n+1}(h)] - \mathbb{E}[\mathbf{E}^{n+1}(h)])\right\| \stackrel{def}{=} T_1 + T_2 + T_3$$

Consider $T_1$, through Property(P1) and $h(s; D, X) \in [h_{min}, h_{max}]$, for all $y \in \mathcal{Y}$, we have

$$\left|E^i(y, h) - \tilde{E}^i(y, h)\right| = \left|\frac{nh_i(y)}{\sum_{j=1, j\neq i}^{n} h_j + h_i(y)} - \frac{h_i(y)}{\mu}\right| \leq h_{max}\left|\frac{n}{\sum_{j=1, j\neq i}^{n} h_j + h_i(y)} - \frac{1}{\mu}\right|$$

$$= h_{max}\frac{\left|\mu - \frac{1}{n}\sum_{j=1}^{n} h_j + \frac{h_i}{n} - \frac{h_i(y)}{n}\right|}{\mu\left(\frac{1}{n}\sum_{j=1, j\neq i}^{n} h_j + \frac{h_i(y)}{n}\right)}$$

$$\leq h_{max}\frac{(h_{max} - h_{min})\sqrt{\frac{log(2/\delta)}{2n}} + \frac{2h_{max}}{n}}{\mu h_{min}} \quad \text{with probability} \geq 1 - \delta$$

For all $y \in \mathcal{Y}$ have the same upper bound, then we have

$$\left\|\mathbf{E}^i - \tilde{\mathbf{E}}^i\right\| = \max_{y \in \mathcal{Y}}\left|E^i(y, h) - \tilde{E}^i(y, h)\right| \leq \frac{h_{max}(h_{max} - h_{min})\sqrt{\frac{log(2/\delta)}{2n}} + \frac{2h_{max}^2}{n}}{\mu h_{min}} \quad \text{with probability} \geq 1 - \delta$$

$$T_1 \leq \frac{L}{n} \sum_{i=1}^{n} \left\| \mathbf{E}^i - \tilde{\mathbf{E}}^i \right\| \leq L \left\| \mathbf{E}^i - \tilde{\mathbf{E}}^i \right\|$$

$$\leq L \frac{h_{max}(h_{max} - h_{min})\sqrt{\frac{log(2/\delta)}{2n}} + \frac{2h_{max}^2}{n}}{\mu h_{min}} \quad \text{with probability} \geq 1 - \delta$$

$$\leq L \frac{h_{max}^2}{h_{min}^2} \left( \sqrt{\frac{log(2/\delta)}{2n}} + \frac{2}{n} \right) \quad \text{with probability} \geq 1 - \delta$$

Consider $T_2$:

$$\mathbb{P}\left( \frac{1}{n} \left\| \sum_{i=1}^{n} (\tilde{\mathbf{E}}^i(h) - \mathbb{E}[\tilde{\mathbf{E}}^{n+1}(h)]) \right\| \geq t \right) = \mathbb{P}\left( \bigcup_{y \in \mathcal{Y}} \left\{ \frac{1}{n} \left| \sum_{i=1}^{n} (\tilde{E}^i(y,h) - \mathbb{E}[\tilde{E}^{n+1}(y,h)]) \right| \geq t \right\} \right)$$

$$\leq \sum_{y \in \mathcal{Y}} \mathbb{P}\left( \frac{1}{n} \left| \sum_{i=1}^{n} (\tilde{E}^i(y,h) - \mathbb{E}[\tilde{E}^{n+1}(y,h)]) \right| \geq t \right)$$

$$\leq \sum_{y \in \mathcal{Y}} \mathbb{P}\left( \frac{1}{n} \left| \sum_{i=1}^{n} \left( \frac{h_i(y)}{\mu} - \mathbb{E}\left[ \frac{h_{n+1}(y)}{\mu^y} \right] \right) \right| \geq t \right)$$

Through the following decomposition and Property P2, there is:

$$\frac{h_i(y)}{\mu} - \mathbb{E}\left[ \frac{h_{n+1}(y)}{\mu^y} \right] = \frac{h_i(y)}{\mu} - \mathbb{E}\left[ \frac{h_{n+1}(y)}{\mu} \right] + \mathbb{E}\left[ \frac{h_{n+1}(y)}{\mu} \right] - \mathbb{E}\left[ \frac{h_{n+1}(y)}{\mu^y} \right]$$

$$\left| \frac{1}{\mu} - \frac{1}{\mu^y} \right| = \left| \frac{\mu^y - \mu}{\mu \mu^y} \right| \leq \frac{h_{max}}{\sqrt{n} h_{min}^2}$$

$$\frac{1}{n} \left| \sum_{i=1}^{n} \left( \frac{h_i(y)}{\mu} - \mathbb{E}\left[ \frac{h_{n+1}(y)}{\mu^y} \right] \right) \right| = \left| \frac{1}{n} \sum_{i=1}^{n} \left( \frac{h_i(y)}{\mu} - \mathbb{E}\left[ \frac{h_{n+1}(y)}{\mu} \right] + \mathbb{E}\left[ \frac{h_{n+1}(y)}{\mu} \right] - \mathbb{E}\left[ \frac{h_{n+1}(y)}{\mu^y} \right] \right) \right|$$

$$= \left| \frac{1}{n} \sum_{i=1}^{n} \left( \frac{h_i(y)}{\mu} - \frac{\mathbb{E}[h_{n+1}(y)]}{\mu} \right) + \mathbb{E}[h_{n+1}(y)] \left( \frac{1}{\mu} - \frac{1}{\mu^y} \right) \right|$$

$$\leq \left| \frac{1}{n} \sum_{i=1}^{n} \left( \frac{h_i(y)}{\mu} - \frac{\mathbb{E}[h_{n+1}(y)]}{\mu} \right) \right| + \frac{h_{max}^2}{\sqrt{n} h_{min}^2}$$

$$\leq \left| \frac{1}{n} \sum_{i=1}^{n} \left( \frac{h_i(y)}{\mu} - \frac{\mathbb{E}[h_i(y)]}{\mu} \right) \right| + \left| \frac{1}{n} \sum_{i=1}^{n} \left( \frac{\mathbb{E}[h_i(y)]}{\mu} - \frac{\mathbb{E}[h_{n+1}(y)]}{\mu} \right) \right| + \frac{h_{max}^2}{\sqrt{n} h_{min}^2}$$

$$\leq \left| \frac{1}{n} \sum_{i=1}^{n} \left( \frac{h_i(y)}{\mu} - \frac{\mathbb{E}[h_i(y)]}{\mu} \right) \right| + \frac{h_{max}}{\sqrt{n} \mu} + \frac{h_{max}^2}{\sqrt{n} h_{min}^2}$$

Return to the derivation of $T_2$, through Property (P1):

$$\mathbb{P}\left( \frac{1}{n} \left| \sum_{i=1}^{n} \left( \frac{h_i(y)}{\mu} - \mathbb{E}\left[ \frac{h_{n+1}(y)}{\mu^y} \right] \right) \right| \geq t \right) \leq \mathbb{P}\left( \left| \frac{1}{n} \sum_{i=1}^{n} \left( \frac{h_i(y)}{\mu} - \frac{\mathbb{E}[h_i(y)]}{\mu} \right) \right| + \frac{h_{max}}{\sqrt{n} \mu} + \frac{h_{max}^2}{\sqrt{n} h_{min}^2} \geq t \right)$$

$$\leq exp\left[ -\frac{2n\mu^2 \left( t - \frac{h_{max}}{\sqrt{n}\mu} - \frac{h_{max}^2}{\sqrt{n}h_{min}^2} \right)^2}{(h_{max} - h_{min})^2} \right]$$

$$\sum_{y \in \mathcal{Y}} \mathbb{P}\left( \frac{1}{n} \left| \sum_{i=1}^{n} \left( \frac{h_i(y)}{\mu} - \mathbb{E}\left[ \frac{h_{n+1}(y)}{\mu^y} \right] \right) \right| \geq t \right) \leq |\mathcal{Y}| exp\left[ -\frac{2n\mu^2 \left( t - \frac{h_{max}}{\sqrt{n}\mu} - \frac{h_{max}^2}{\sqrt{n}h_{min}^2} \right)^2}{(h_{max} - h_{min})^2} \right]$$

Let the right end of the inequality be $\delta$, and inverse solution $t$:

$$
\begin{aligned}
T_2 &= \frac{L}{n} \left\| \sum_{i=1}^{n} \left( \tilde{\mathbf{E}}^i(h) - \mathbb{E}[\tilde{\mathbf{E}}^{n+1}(h)] \right) \right\| \\
&\leq L \left( \frac{h_{max} - h_{min}}{\mu} \sqrt{\frac{log(2|\mathcal{Y}|/\delta)}{2n}} + \frac{h_{max}}{\sqrt{n}\mu} + \frac{h_{max}^2}{\sqrt{n}h_{min}^2} \right) \quad \text{with probability} \geq 1 - \delta \\
&\leq L \frac{h_{max}^2}{h_{min}^2} \left( \sqrt{\frac{log(2|\mathcal{Y}|/\delta)}{2n}} + \frac{2}{\sqrt{n}} \right) \quad \text{with probability} \geq 1 - \delta
\end{aligned}
$$

Consider $T_3$, for all $y \in \mathcal{Y}$:

$$
\begin{aligned}
\left| \tilde{E}^{n+1}(y, h) - E^{n+1}(y, h) \right| &= \left| \frac{h_{n+1}(y)}{\mu^y} - \frac{(n+1)h_{n+1}(y)}{\sum_{i=1}^{n} h_i^y + h_{n+1}(y)} \right| \leq h_{max} \frac{\left| \mu^y - \frac{1}{n+1}\sum_{i=1}^{n} h_i^y - \frac{1}{n+1}h_{n+1}(y) \right|}{\mu^y \left( \frac{1}{n+1} \left( \sum_{i=1}^{n} h_i^y + h_{n+1}(y) \right) \right)} \\
&\leq h_{max} \frac{\left| \mu^y - \frac{1}{n+1}\sum_{i=1}^{n} h_i^y \right| + \frac{h_{max}}{n+1}}{h_{min}\mu^y} \leq h_{max} \frac{\frac{n}{n+1} \left| \mu^y - \frac{1}{n}\sum_{i=1}^{n} h_i^y \right| + \frac{nh_{max}}{(n+1)^2} + \frac{h_{max}}{n+1}}{h_{min}\mu^y} \\
&\leq h_{max} \frac{\frac{n(h_{max}-h_{min})}{n+1} \sqrt{\frac{log(2/\delta)}{2n}} + \frac{nh_{max}}{(n+1)^2} + \frac{h_{max}}{n+1}}{h_{min}\mu^y} \quad \text{with probability} \geq 1 - \delta \\
&\leq \frac{h_{max}^2}{h_{min}^2} \left( \sqrt{\frac{log(2/\delta)}{2n}} + \frac{2}{n} \right) \quad \text{with probability} \geq 1 - \delta
\end{aligned}
$$

This is the upper bound for all $y \in \mathcal{Y}$, and thus it is the same for the upper bound of the norm. Let the right end of the inequality be $t$, and inverse solution $\delta$:

$$
\mathbb{P} \left( \left\| \tilde{\mathbf{E}}^{n+1}(h) - \mathbf{E}^{n+1}(h) \right\| \geq t \right) \leq 2exp \left[ -2n \left( \frac{h_{min}^2}{h_{max}^2}t - \frac{2}{n} \right)^2 \right]
$$

$$
\begin{aligned}
\mathbb{E} \left[ \left| \tilde{\mathbf{E}}^{n+1}(h) - \mathbf{E}^{n+1}(h) \right| \right] &= \int_0^{\infty} \mathbb{P} \left( \left\| \tilde{\mathbf{E}}^{n+1}(h) - \mathbf{E}^{n+1}(h) \right\| \geq t \right) dt \leq \int_0^{\infty} 2exp \left[ -2n \left( \frac{h_{min}^2}{h_{max}^2}t - \frac{2}{n} \right)^2 \right] dt \\
&\leq \int_{-\frac{\sqrt{8}}{\sqrt{n}}}^{\infty} \frac{\sqrt{2}h_{max}^2}{\sqrt{n}h_{min}^2} exp \left[ -s^2 \right] ds \leq \frac{\sqrt{2}h_{max}^2}{\sqrt{n}h_{min}^2} \int_{-\infty}^{\infty} exp \left[ -s^2 \right] ds \\
&= \frac{\sqrt{2\pi}h_{max}^2}{\sqrt{n}h_{min}^2}
\end{aligned}
$$

Return to the derivation of $T_3$:

$$
\begin{aligned}
T_3 &= \frac{L}{n} \left\| \sum_{i=1}^{n} \left( \mathbb{E}[\tilde{\mathbf{E}}^{n+1}(h)] - \mathbb{E}[\mathbf{E}^{n+1}] \right) \right\| = L \left\| \mathbb{E}[\tilde{\mathbf{E}}^{n+1}(h)] - \mathbb{E}[\mathbf{E}^{n+1}(h)] \right\| \\
&\leq L\mathbb{E} \left[ \left\| \tilde{\mathbf{E}}^{n+1}(h) - \mathbf{E}^{n+1}(h) \right\| \right] \leq \frac{\sqrt{2\pi}Lh_{max}^2}{\sqrt{n}h_{min}^2}
\end{aligned}
$$

Finally, we have:

$$
|\hat{\alpha}^{LOO}(h) - \mathbb{E}[\tilde{\alpha}_{n+1}(h)]| \leq L \frac{h_{max}^2}{h_{min}^2} \left( \sqrt{\frac{log(2/\delta)}{2n}} + \sqrt{\frac{log(2|\mathcal{Y}|/\delta)}{2n}} + \sqrt{\frac{2\pi}{n}} + \frac{4}{n} \right) \quad \text{with probability} \geq 1 - 2\delta
$$

It means that:

$$
|\hat{\alpha}^{LOO}(h) - \mathbb{E}[\tilde{\alpha}_{n+1}(h)]| = O_P(\frac{1}{\sqrt{n}})
$$

$\square$

# G. Proof of Theorem 3.2

**Theorem 3.2.** *Assume that the transformation* $\forall h(s; D, X) \in \mathcal{H}$ *further satisfies the following property:*

*(P4) For all $y \in \mathcal{Y}$ have:*

$$\lim_{n \to \infty} \left( \frac{\sum_{i=1}^{n} h_i}{n} - \frac{\sum_{i=1}^{n} h_i^y}{n} \right) = 0$$

*Then,* $\forall h^1, h^2 \in \mathcal{H}$, *when $n$ is sufficiently large, we have:*

$$C_{h^1}^{\tilde{\alpha}_{n+1}(h^1)}(X_{n+1}) = C_{h^2}^{\tilde{\alpha}_{n+1}(h^2)}(X_{n+1}) \tag{23}$$

Property (P4) implies that adding a single pseudo-test point has an asymptotically negligible effect on the average transformed calibration scores. Since the transformation depends only on the statistics of the input dataset $D$ and feature $X$, the perturbation induced by a single additional point in $D$ vanishes as the calibration set size becomes sufficiently large.

Moreover, if $h_i = h_i^y$ holds, then Property (P4) is automatically satisfied, and the theorem no longer requires any assumption on the calibration set size. Beyond the setting described in Remark 3.3, the condition $h_i = h_i^y$ may also arise in other scenarios. For example, it holds when the transformation depends only on highly localized statistics (e.g., nearest neighbors within a fixed radius) and the pseudo-test point does not belong to the local neighborhood of any calibration sample. In addition, if the underlying statistics are sufficiently robust (e.g., quantiles) such that the inclusion of a pseudo-test point does not alter these statistics, then $h_i = h_i^y$ also remains valid.

*Proof.* Through Property (P4), $\forall \epsilon > 0$, $\exists N \in \mathbb{N}^+$, $\forall n \geq N$, we have:

$$\frac{\sum_{i=1}^{n} h_i}{n} - \epsilon < \frac{\sum_{i=1}^{n} h_i^y}{n} < \frac{\sum_{i=1}^{n} h_i}{n} + \epsilon$$

We take $\epsilon$ as sufficiently small

$$\forall h \in \mathcal{H}, C_h^\alpha(X_{n+1}) = \left\{ y : E^{n+1}(y, h) < \frac{1}{\alpha} \right\} = \left\{ y : h_{n+1}(y) < \frac{1}{(n+1)\alpha - 1} \sum_{i=1}^{n} h_i^y \right\}$$

$$= \left\{ y : S(X_{n+1}, y) < h^{-1} \left( \frac{1}{(1/n+1)\alpha - 1/n} \frac{\sum_{i=1}^{n} h_i^y}{n}; D_{n+1}, X_{n+1} \right) \right\}$$

$$= \left\{ y : S(X_{n+1}, y) < h^{-1} \left( \frac{1}{(1/n+1)\alpha - 1/n} \frac{\sum_{i=1}^{n} h_i}{n}; D_{n+1}, X_{n+1} \right) \right\}$$

Similarly:

$$\forall h \in \mathcal{H}, C_h^\alpha(X_i) = \left\{ y : S(X_i, y) < h^{-1} \left( \frac{1}{n\alpha - 1} \sum_{j=1, j \neq i}^{n} h_j; D_i, X_i \right) \right\} \quad i = 1, \cdots, n$$

Define

$$w(D, X) = \sup\{l > 0 : |\{y : S(X, y) < l\}| \leq \mathcal{T}(D, X)\}$$

Which converts the size constraint of the prediction set into a score constraint. Then:

$$|C_h^\alpha(X_{n+1})| \leq \mathcal{T}(D_{n+1}, X_{n+1}) \Leftrightarrow h^{-1} \left( \frac{1}{(n+1)\alpha - 1} \sum_{i=1}^{n} h_i; D_{n+1}, X_{n+1} \right) \leq w(D_{n+1}, X_{n+1})$$

$$\Leftrightarrow \frac{1}{(n+1)\alpha - 1} \sum_{i=1}^{n} h_i \leq h(w(D_{n+1}, X_{n+1}); D_{n+1}, X_{n+1})$$

$$\Leftrightarrow \alpha \geq \frac{1}{n+1} \left[ \frac{\sum_{i=1}^{n} h_i}{h\left(w(D_{n+1}, X_{n+1})\right)} + 1 \right]$$

$$|C_h^\alpha(X_i)| \leq \mathcal{T}(D_i, X_i) \Leftrightarrow \alpha \geq \frac{1}{n}\left[\frac{\sum_{j=1, j\neq i} h_j}{h\left(w(D_i, X_i)\right)} + 1\right], \quad i = 1, \cdots, n$$

From the definition of $\tilde{\alpha}(h)$:

$$\tilde{\alpha}_{n+1}(h) = inf\{\alpha \in (0,1) : |C_h^\alpha(X_{n+1})| \leq \mathcal{T}(D_{n+1}, X_{n+1})\} = \frac{1}{n+1}\left[\frac{\sum_{i=1}^n h_i}{h\left(w(D_{n+1}, X_{n+1})\right)} + 1\right]$$

$$\tilde{\alpha}_i(h) = inf\{\alpha \in (0,1) : |C_h^\alpha(X_i)| \leq \mathcal{T}(D_i, X_i)\} = \frac{1}{n}\left[\frac{\sum_{j=1, j\neq i} h_j}{h\left(w(D_i, X_i)\right)} + 1\right], \quad i = 1, \cdots, n$$

This is precisely the result of the Corollary(3.4). Substitute $\tilde{\alpha}_{n+1}(h)$ into the calculation formula of the prediction set $C_h^{\tilde{\alpha}_{n+1}(h)}(X_{n+1})$.

$$C_h^{\tilde{\alpha}_{n+1}(h)}(X_{n+1}) = \{y : h_{n+1}(y) < h(w(D_{n+1}, X_{n+1}); D_{n+1}, X_{n+1})\} = \{y : S(X_{n+1}, y) < w(D_{n+1}, X_{n+1})\}$$

The prediction set is independent of $h$. So:

$$C_{h^1}^{\tilde{\alpha}_{n+1}(h^1)}(X_{n+1}) = C_{h^2}^{\tilde{\alpha}_{n+1}(h^2)}(X_{n+1}) \quad \forall h^1, h^2 \in \mathcal{H}$$

$\square$

# H. Proof of Theorem 3.5 and

**Theorem 3.5.** *Consider the simplified setting where:*

*(P5)*

$$\mathbb{E}\left[\frac{h_i}{h(w_{n+1}; D_{n+1}, X_{n+1})}\right] = \mathbb{E}[h_i]\mathbb{E}\left[\frac{1}{h(w_{n+1}; D_{n+1}, X_{n+1})}\right]$$

*(P6)*

$$\exists c_0 > 0, h(w_{n+1}; D_{n+1}, X_{n+1}) > c_0 \quad a.s$$

*Then we have:*

$$\frac{a\mathbb{I}(s \geq w(D, X))}{\sqrt{\mathbb{P}(S(X, Y) \geq w(D, X)|D, X)}} = \underset{h \in \bar{\mathcal{H}}}{argmin} : \mathbb{E}\left[\frac{h_i}{h(w_{n+1}; D_{n+1}, X_{n+1})}\right]$$

*where $a$ is an arbitrary positive constant.*

*Proof.* $\forall h(s; D, X) \in \bar{\mathcal{H}}$, define $h^*(s; D, X) = \mathbb{I}(s \geq w(D, X))h(s; D, X)$. Clearly, $h^*(s; D, X) \in \bar{\mathcal{H}}$

$$\mathbb{E}[h_i] = \mathbb{E}\left[h\left(S(X_i, Y_i); D_i, X_i\right)\right] = \mathbb{E}\left[\mathbb{E}\left[h\left(S(X_i, Y_i); D_i, X_i\right)|D_i, X_i\right]\right]$$

$$= \mathbb{E}\left[\mathbb{E}\left[\sum_{S(X_i, y) \geq w_i} h(S(X_i, y); D_i, X_i)\mathbb{P}_{Y|X_i}(Y = y) + \sum_{S(X_i, y) < w_i} h(S(X_i, y); D_i, X_i))\mathbb{P}_{Y|X_i}(Y = y)\Big| D_i, X_i\right]\right]$$

$$\geq \mathbb{E}\left[\mathbb{E}\left[\sum_{S(X_i, y) \geq w_i} h(S(X_i, y); D_i, X_i)\mathbb{P}_{Y|X_i}(Y = y)\Big| D_i, X_i\right]\right]$$

$$= \mathbb{E}\left[\mathbb{E}\left[\sum_{S(X_i, y) \geq w_i} h^*(S(X_i, y); D_i, X_i)\mathbb{P}_{Y|X_i}(Y = y) + \sum_{S(X_i, y) < w_i} h^*(S(X_i, y); D_i, X_i)\mathbb{P}_{Y|X_i}(Y = y)\Big| D_i, X_i\right]\right]$$

$$= \mathbb{E}[h_i^*]$$

$$\mathbb{E}\left[\frac{1}{h(w_{n+1}; D_{n+1}, X_{n+1})}\right] = \mathbb{E}\left[\frac{\mathbb{I}(w_{n+1} \geq w_{n+1})}{h(w_{n+1}; D_{n+1}, X_{n+1})}\right] = \mathbb{E}\left[\frac{1}{h^*(w_{n+1}; D_{n+1}, X_{n+1})}\right]$$

Then:

$$\mathbb{E}[h_i]\mathbb{E}\left[\frac{1}{h(w_{n+1}; D_{n+1}, X_{n+1})}\right] \geq \mathbb{E}[h_i^*]\mathbb{E}\left[\frac{1}{h^*(w_{n+1}; D_{n+1}, X_{n+1})}\right]$$

So we only need to focus on the part where $s \geq w(D, X)$.

Define $c(D, x) = h^*(w(D, X); D, X)$, and $|c(D, x)| \geq c_0 > 0$ by Property(P6). Let $c_i = c(D_i, X_i)$   $i = 1, \cdots, n+1$

$$\mathbb{E}[h_i^*]\mathbb{E}\left[\frac{1}{h^*(w_{n+1}; D_{n+1}, X_{n+1})}\right]$$

$$= \mathbb{E}\left[\mathbb{E}\left[\sum_{S(X_i, y) \geq w_i} h(S(X_i, y); D_i, X_i)\mathbb{P}_{Y|X_i}(Y = y)\Big| D_i, X_i\right]\right]\mathbb{E}\left[\frac{1}{c_{n+1}}\right]$$

$$\geq \mathbb{E}\left[\mathbb{E}\left[\sum_{S(X_i, y) \geq w_i} c_i\mathbb{P}_{Y|X_i}(Y = y)\Big| D_i, X_i\right]\right]\mathbb{E}\left[\frac{1}{c_{n+1}}\right]$$

The equality sign holds if and only if: $h(s; D, X) = c(D, X)$, when $s \geq w(D, X)$.

So we only need to consider the optimal $c(D, X)$. Define $h^c(s; D, X) = \mathbb{I}(s \geq w(D, X))c(D, X)$ and

$$p(D, X) = \sum_{S(X, y) \geq w(D, X)} \mathbb{P}_{Y|X}(Y = y), \quad p_i = p(D_i, X_i), i = 1, \cdots, n+1$$

Then:

$$\mathbb{E}[h_i^c] = \mathbb{E}\left[\mathbb{E}\left[c_i p_i\Big| D_i, X_i\right]\right] = \int_{\mathcal{D}\times\mathcal{X}} c(D, X)p(D, X)d\mathcal{P}(D_i, X_i) \overset{\text{def}}{=} A(c)$$

$$\mathbb{E}\left[\frac{1}{h^c(w_{n+1}; D_{n+1}, X_{n+1})}\right] = \mathbb{E}\left[\frac{1}{c_{n+1}}\right] = \int_{\mathcal{D}\times\mathcal{X}} \frac{1}{c(D, X)}d\mathcal{P}(D_{n+1}, X_{n+1}) \overset{\text{def}}{=} B(c)$$

Consider a small perturbation of $c(D, X)$ given by $c_\epsilon(D, X) = c(D, X) + \epsilon\,\eta(D, X)$, where $\eta(D, X)$ is a fixed bounded function satisfying $|\eta(D, X)| \leq M$ and $\epsilon > 0$ is sufficiently small with $\epsilon < \frac{c_0}{2M}$. Then the corresponding variations are:

$$\delta A(c) = A(c_\epsilon) - A(c) = \int_{\mathcal{D}\times\mathcal{X}} \epsilon\eta(D, X)p(D, X)d\mathcal{P}(D_i, X_i)$$

$$\delta B(c) = B(c_\epsilon) - B(c) = \int_{\mathcal{D}\times\mathcal{X}} \frac{1}{c(D, X) + \epsilon\eta(D, X)} - \frac{1}{c(D, X)}d\mathcal{P}(D_{n+1}, X_{n+1})$$

Next, we will derive the linear principal part of the variation in $\delta B(c)$

$$\frac{1}{1+u} = \sum_{k=0}^{\infty}(-1)^k u^k \xrightarrow{0 < \epsilon < \frac{c_0}{2M}} \frac{1}{1 + \epsilon\frac{\eta(D,X)}{c(D,X)}} = \sum_{k=0}^{\infty}(-1)^k\left(\epsilon\frac{\eta(D,X)}{c(D,X)}\right)^k$$

$$\frac{1}{c_\epsilon(D, X)} - \frac{1}{c(D, X)} = \frac{1}{c(D, X)}\left(\frac{1}{1 + \epsilon\frac{\eta(D,X)}{c(D,X)}} - 1\right) = -\frac{\epsilon\eta(D, X)}{c(D, X)^2} + \sum_{k=2}^{\infty}(-1)^k\frac{\eta(D, X)^k}{c(D, X)^{k+1}}\epsilon^k$$

$$\left|\sum_{k=2}^{\infty}(-1)^k\frac{\eta(D, X)^k}{c(D, X)^{k+1}}\epsilon^k\right| \leq \sum_{k=2}^{\infty}\frac{M^k\epsilon^k}{c_0^{k+1}} = \frac{M^2}{c_0^2(c_0 - M\epsilon)}\epsilon^2$$

This is a consistent upper bound:

$$\left| \int_{\mathcal{D} \times \mathcal{X}} \sum_{k=2}^{\infty} (-1)^k \frac{\eta(D,X)^k}{c(D,X))^{k+1}} \epsilon^k d\mathcal{P}(D_{n+1}, X_{n+1}) \right| \leq \int_{\mathcal{D} \times \mathcal{X}} \left| \sum_{k=2}^{\infty} (-1)^k \frac{\eta(D,X)^k}{c(D,X)^{k+1}} \epsilon^k \right| d\mathcal{P}(D_{n+1}, X_{n+1})$$

$$\leq \int_{\mathcal{D} \times \mathcal{X}} \frac{M^2}{c_0^2(c_0 - M\epsilon)} \epsilon^2 d\mathcal{P}(D_{n+1}, X_{n+1})$$

$$= \frac{M^2}{c_0^2(c_0 - M\epsilon)} \epsilon^2 \int_{\mathcal{D} \times \mathcal{X}} d\mathcal{P}(D_{n+1}, X_{n+1})$$

$$= \frac{M^2}{c_0^2(c_0 - M\epsilon)} \epsilon^2 = O(\epsilon^2)$$

Then:

$$\delta B(c) = \int_{\mathcal{D} \times \mathcal{X}} -\frac{\epsilon \eta(D,X)}{c(D,X)^2} d\mathcal{P}(D_{n+1}, X_{n+1}) + O(\epsilon^2)$$

$O(\epsilon^2)$ can be ignored in the first-order variation. We define $J(c) = A(c)B(c)$, its variation is:

$$\delta J(c) = A(c)\delta B(c) + B(c)\delta A(c) = \epsilon \int_{\mathcal{D} \times \mathcal{X}} B(c)p(D,X) - \frac{A(c)}{c(D,X)^2} d\mathcal{P}(D_{n+1}, X_{n+1})$$

$$\forall |\eta(D,X)| < M, J' = \frac{\delta J}{\epsilon} = 0 \Rightarrow B(c)p(D,X) - \frac{A(c)}{c(D,X)^2} = 0 \Rightarrow c(D,X) = \sqrt{\frac{A(c)}{B(c)p(D,X)}}$$

This is the first-order condition. Under this condition, continue to search for $c(D,X)$. Pass $c(D,X)$ back to $J(c)$:

$$J(c) = A(c)B(c) = c(D,X)\mathbb{E}\left[\sqrt{p(D,X)}\right] \frac{1}{c(D,X)} \mathbb{E}\left[\sqrt{p(D,X)}\right] = \left(\mathbb{E}\left[\sqrt{p(D,X)}\right]\right)^2$$

At this point, $J(c)$ is independent of $c(D,X)$. It means that $\sqrt{\frac{A(c)}{B(c)}}$ can be an arbitrary positive constant.

$$c(D,X) = a\sqrt{\frac{1}{p(D,X)}} = argmin : J(c)$$

$$h(s; D, X) = \frac{a\mathbb{I}(s > w(D,X))}{\sqrt{p(D,X)}} = \underset{h \in \bar{H}}{argmin} : \mathbb{E}[h_i]\mathbb{E}\left[\frac{1}{h(w_{n+1}; D_{n+1}, X_{n+1})}\right]$$

$a$ is an arbitrary positive constant. By Property(P5), we thus proved the theorem. $\qquad \square$

## I. Approximation of Strictly Monotonic Functions

**Proposition I.1.** *Suppose $0 < w_m \leq w(D,X) \leq w_M$, then $\forall w_m/2 > \epsilon > 0$, exists $\mathbb{I}_w^\epsilon \in \mathcal{H}$, such that:*

$$||\mathbb{I}_w - \mathbb{I}_w^\epsilon|| \leq \epsilon$$

$$\left|\hat{\alpha}^{LOO}(\mathbb{I}_w^\epsilon) - \hat{\alpha}_{pe}^{LOO}(\mathbb{I}_w)\right| \leq \frac{2(w_m + (n-1)w_M)\epsilon}{nw_m^2}$$

$\hat{\alpha}_{pe}^{LOO}$ *is calculated based on Corollary 3.4*

This proposition shows that the optimization remains valid even when the constraints are expanded from $\mathcal{H}$ to $\bar{\mathcal{H}}$. For a sufficiently small $\epsilon$, the difference in the leave-one-out estimators between the strictly increasing $\mathbb{I}_w^\epsilon$ and the non-decreasing $\mathbb{I}_w$ is also sufficiently small. While $\mathbb{I}_w$ is convenient to compute, its miscoverage level cannot be determined by the original definition. Instead, the calculation must follow the expression derived under the monotonicity condition.

In this Proposition, the norm is defined as $||h|| = \underset{s \in [0, \infty)}{sup} |h(s)|$

*Proof.* For transformation $h(s; D, X) = w(D, X)\mathbb{I}(s > w(D, X)) = \mathbb{I}_w$, we fixed $D, X$. We can add a weak partial strictly monotonically increasing function to make it satisfy strict monotonicity in different parts. For $\mathbb{I}(s > w(D, X))$, we just consider two parts: $\{s : s < w(D, X)\}$ and $\{s : s \geq w(D, X)\}$. We select the function $\Delta^\epsilon(s; w(D, X))$: as the weak partial strictly monotonically increasing function:

$$\Delta^\epsilon(s; w(D, X)) = \mathbb{I}(s < w(D, X))\frac{\epsilon}{w(D, X)}s + \mathbb{I}(s \geq w(D, X))\frac{-\epsilon}{s - w(D, X) + 1}$$

And define:

$$\mathbb{I}_w^\epsilon = \mathbb{I}_w + \Delta^\epsilon(s; w(D, X)) = w(D, X)\mathbb{I}(s > w(D, X)) + \Delta^\epsilon(s; w(D, X))$$

Its function graph is

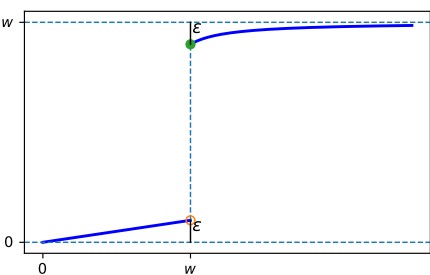

Clearly, $\forall s \geq 0 \quad |\Delta^\epsilon(s; w(D, X))| \leq \epsilon \quad$ and $\quad \|\mathbb{I}_w^\epsilon - \mathbb{I}^\epsilon\| = \|\Delta^\epsilon(s; w(D, X))\| \leq \epsilon$

Through Corollary(3.4):

$$\tilde{\alpha}_{n+1}(\mathbb{I}_w^\epsilon) = \frac{1}{n+1}\left(\frac{\sum_{i=1}^n [w_i\mathbb{I}(S_i \geq w_i) + \Delta^\epsilon(S_i; w_i)]}{w_{n+1} - \epsilon} + 1\right)$$

$$\tilde{\alpha}_{n+1}^{pe}(\mathbb{I}_w) = \frac{1}{n+1}\left(\frac{\sum_{i=1}^n w_i\mathbb{I}(S_i \geq w_i)}{w_{n+1}} + 1\right)$$

$$\tilde{\alpha}_i(\mathbb{I}_w^\epsilon) = \frac{1}{n}\left(\frac{\sum_{j=1, j\neq i}^n [w_j\mathbb{I}(S_j \geq w_j) + \Delta^\epsilon(S_j; w_j)]}{w_i - \epsilon} + 1\right)$$

$$\tilde{\alpha}_i^{pe}(\mathbb{I}_w) = \frac{1}{n}\left(\frac{\sum_{j=1, j\neq i}^n w_j\mathbb{I}(S_j \geq w_j)}{w_i} + 1\right)$$

$$\hat{\alpha}_{pe}^{LOO}(h) \stackrel{\text{def}}{=} \frac{1}{n}\sum_{i=1}^n \tilde{\alpha}_i^{pe}(h)$$

Since the two functions are very close, intuitively, the corresponding levels of false coverage are also very close. Under $\|\Delta^\epsilon(s; w(D, X))\| \leq \epsilon$ always holding for all $(D, X) \in \mathcal{D} \times \mathcal{X}$, we have:

$$\left|\tilde{\alpha}_{n+1}(\mathbb{I}_w^\epsilon) - \tilde{\alpha}_{n+1}^{pe}(\mathbb{I}_w)\right| = \frac{1}{n+1}\left[\left(\frac{1}{w_{n+1} - \epsilon} - \frac{1}{w_{n+1}}\right)\sum_{i=1}^n w_i\mathbb{I}(S_i \geq w_i) + \frac{1}{w_{n+1} - \epsilon}\sum_{i=1}^n |\Delta^\epsilon(S_i; D_i, X_i)|\right]$$

$$\leq \frac{1}{n+1}\left[\left(\frac{1}{w_{n+1} - \epsilon} - \frac{1}{w_{n+1}}\right)w_M n + \frac{n\epsilon}{w_{n+1} - \epsilon}\right] \leq \frac{2(w_m + nw_M)\epsilon}{(n+1)w_m^2}$$

Similarly, for leave-one-out estimator and pseudo miscoverage levels:

$$\left|\tilde{\alpha}_i(\mathbb{I}_w^\epsilon) - \tilde{\alpha}_i^{pe}(\mathbb{I}_w)\right| \leq \frac{2(w_m + (n-1)w_M)\epsilon}{nw_m^2}$$

$$\left|\hat{\alpha}^{LOO}(\mathbb{I}_w^\epsilon) - \hat{\alpha}_{pe}^{LOO}(\mathbb{I}_w)\right| \leq \frac{2(w_m + (n-1)w_M)\epsilon}{nw_m^2}$$

$\square$

