# OpenReview forum: "Improving Backward Conformal Prediction via Non-Conformity Score Transformation"
_ICML.cc/2026/Conference — ICML 2026 regular_

### Official Review · Reviewer_H4sy · 2026-02-25

**Soundness:** 3
**Presentation:** 4
**Significance:** 3
**Originality:** 4
**Overall Recommendation:** 5
**Confidence:** 3

**Summary:**

The paper proposes ST-BCP, a method that applies a data-dependent transformation to non-conformity scores. The core idea is to reshape the score distribution to maximize the tightness of Markov's inequality (effectively pushing the distribution towards a two-point distribution). The authors introduce a symmetric parameterization strategy to preserve exchangeability and derive a theoretically optimal transformation (a step function) under monotonicity constraints. Experiments on CIFAR-10, CIFAR-100, and Tiny-ImageNet show that ST-BCP significantly reduces the gap between the estimated coverage bound and empirical coverage compared to the baseline BCP.

**Compliance With Llm Reviewing Policy:**

Affirmed.

**Final Justification:**

The authors propose ST-BCP, which represents a meaningful contribution to conformal prediction and can remedy the coverage gap that arises in backward conformal prediction. I recommend acceptance.

**Key Questions For Authors:**

* **Computational Cost:** Could you provide a wall-clock time comparison between standard BCP and ST-BCP during inference? specifically, how does the complexity scale with the number of classes $|\mathcal{Y}|$ given the nested loops in Algorithm 1?
 * **Taylor Approximation:** In Appendix D, you mention that in low-coverage regimes, the guarantee may not hold. Does the proposed step-function transformation ($I_w$) exacerbate or mitigate the error of the Taylor approximation compared to the identity transformation?
* **Hyperparameters:** The method introduces $w(D, X)$ which relies on calculating $\mathcal{T}$. For the feature-dependent constraints, how sensitive is the method to the hyperparameters used in $\mathcal{T}$ (e.g., $k$ in k-NN)?

**Limitations:**

Yes

**Strengths And Weaknesses:**

* **Theoretical Novelty and Rigor:** The derivation of the optimal transformation is mathematically elegant. The insight that Markov's inequality is tightest for a two-point distribution, and subsequently designing a transformation ($I_w$) to mimic this behavior, is a strong theoretical contribution. Furthermore, the use of symmetric parameterization to ensure that data-dependent transformations do not violate the exchangeability assumption is critical and well-handled.
* **Significant Empirical Improvement:** The results are compelling. Reducing the coverage gap (GAP) from ~4.2% to ~1.1% (average) while maintaining or improving the Mean Squared Error (MSE) of the estimator demonstrates that the method is not just a theoretical exercise but has practical utility. The method makes the BCP bound much tighter and more informative.
* **Comprehensive Analysis of the LOO Estimator:** The paper provides a solid consistency proof for the Leave-One-Out (LOO) estimator under the transformed scores. The analysis of the estimator's concentration (Figure 1) provides good intuition on why the transformation works—it sharply reduces the variance of the estimator.

* **Computational Complexity:** Algorithm 1 suggests a high computational cost. The nested loops (Line 8-15) imply that for every test point, and for every candidate label $y \in \mathcal{Y}$, the algorithm iterates through $n$ calibration points to compute $T(D_i^y, X_i)$ and $w(D_i^y, X_i)$. If the size constraint rule $\mathcal{T}$ involves complex operations (like k-NN or PCA as mentioned in Appendix B), doing this $O(|\mathcal{Y}| \cdot n)$ times for each inference seems prohibitively expensive for real-time applications with large label spaces (e.g., ImageNet with 1000 classes). The paper claims "asymptotic efficiency," but the wall-clock time impact should be discussed.

* **Dependence on $\mathcal{T}$ Definition:** The optimization of the transformation depends heavily on the threshold $w(D, X)$, which in turn depends on the size constraint $\mathcal{T}$. If $\mathcal{T}$ is noisy or unstable (e.g., based on local k-NN entropy), the transformation $I_w$ becomes a step function at a noisy threshold. The impact of the stability of $\mathcal{T}$ on the stability of the coverage bound is not fully explored.
* **Other question:** A fundamental limitation of the BCP framework (and by extension, this work) is the reliance on a first-order Taylor approximation (Eq. 7 and Appendix D) to translate the expectation of the e-variable to a probability bound. The authors acknowledge in Appendix D that this approximation can fail in low-coverage regimes. While the proposed transformation tightens the bound, it does not inherently fix the validity issue of the Taylor approximation itself. If the approximation error is large, the "guarantee" is not rigorous. The paper would benefit from a more quantitative analysis of when this approximation breaks down under their specific transformation.

---

> ### Author Rebuttal · Authors · 2026-03-31
>
> Thank you for the valuable comments. Please find our response below.
>
> 1. **Measured wall-clock time for single inference step** [W1,Q1]
>
> We measured the average time for a single inference step ($n$=200, caching feature extraction, averaged over 10 runs):
>
> | Dataset | $\vert\mathcal{Y}\vert$ | BCP (s) | ST-BCP (s) |
> | :--- | :---: | :---: | :---: |
> | CIFAR-10 | 10 | 0.087 | 1.12 |
> | CIFAR-100 | 100 | 0.093 | 10.4 |
> | TinyImageNet | 200 | 0.098 | 21.1 |
>
> As shown, the serial execution time indeed scales linearly with the label space. However, we would like to highlight that ST-BCP is primarily designed for risk-sensitive scenarios such as the medical diagnostic systems mentioned in our response to Reviewer PLDF(Q2). In such domains, accepting an inference latency is highly worthwhile for achieving a tighter coverage bound that prevents unwarranted human intervention. We will explicitly discuss this trade-off between computational cost and a tighter coverage bound in the final version.
>
> 2. **The influence of $\mathcal{T}$'s noise on the coverage bound** [W2]
>
> We clarify that the noise of $\mathcal{T}$ has a very small impact on the coverage bound, and we will explain this theoretically.
> In ST-BCP, the practically reported coverage bound is the LOO estimator $\hat{\alpha}^{LOO}$. We fully agree with your premise: if $\mathcal{T}$ is noisy, it causes the threshold $w$ to fluctuate accordingly, resulting in a "noisy threshold." Since our transformation function $\mathbb{I}_w$ depends directly on $w$, analyzing the impact of $w$'s fluctuations on the LOO estimator $\hat{\alpha}^{LOO}$ directly and equivalently answers how the noise in $\mathcal{T}$ affects the coverage bound.
>
> $$\hat{\alpha}^{LOO} =\frac{1}{n^2}\left(\sum_{i=1}^nw_i\mathbb{I}(S_i\geq w_i)\right)\left(\sum_{i=1}^n\frac{1}{w_i}\right)-\frac{1}{n^2}\sum_{i=1}^n\mathbb{I}(S_i\geq w_i)+\frac{1}{n}$$
>
> It can be clearly seen from this formula that even if the noise in $\mathcal{T}$causes fluctuations in $w$, these fluctuations will be reduced during the averaging process. Therefore, our method can still output a stable coverage bound even when there is noise in $\mathcal{T}$.
>
> 3. **Does the transformation we proposed affect the Taylor error？** [W3,Q2]
>
> NO, we provide a theoretical clarification to confirm that our proposed transformation $\mathbb{I}_w$ does not affect the Taylor error. The primary factor influencing the Taylor error is the concentration of $\tilde{\alpha}(h)$. When $\tilde{\alpha}(h)$ is highly concentrated around its expectation, the higher-order central moments decay, and the Taylor error smoothly approaches 0. We formally prove this by expanding the approximated term. The exact first-order Taylor expansion holds if and only if:
>
> $$\mathbb{E}\left[\frac{A}{B}\right]= \frac{\mathbb{E}[A]}{\mathbb{E}[B]} \Leftrightarrow \frac{1}{\mathbb{E}[B]} \left[ \sum_{k=1}^{\infty} (-1)^k \frac{\mathbb{E}[A(B-\mathbb{E}[B])^k]}{(\mathbb{E}[B])^k} \right]=0$$where$$A=\mathbb{P}\left(Y\notin C_{h}^{\tilde{\alpha}(h)}\left(X\right)\mid\tilde{\alpha}(h)\right),B=\tilde{\alpha}(h)$$
>
> Building on this insight, we further analyze why our score transformation $\mathbb{I}_w$ strictly preserves this concentration. The analytical expression for $\tilde{\alpha}(h)$ is:
>
> $$\tilde{\alpha}(h)=\frac{(\sum_{i=1}^n{h_i})/(n+1)}{h(w_{n+1};D_{n+1},X_{n+1})}+\frac{1}{n+1}$$
>
> Given that the numerator (an empirical mean) is highly concentrated, the lack of concentration in $\tilde{\alpha}(h)$ primarily stems from the denominator. Remarkably, whether evaluating the baseline BCP(identity transformation $h(s;D,X)=s$) or our proposed $\mathbb{I}_w:h(s;D,X)=w(D,X)\mathbb{I}(s\geq w(D,X))$, this specific denominator yields the exact same value:
>
> $$h(w_{n+1};D_{n+1},X_{n+1})=w_{n+1} \quad (\text{for both } \mathbb{I}_w \text{ and identity transformation})$$
>
> Therefore, our transformation $\mathbb{I}_w$ does not alter the concentration of $\tilde{\alpha}(h)$. It is specifically designed to tighten the Markov bound and does not exacerbate or mitigate the Taylor error.
>
> 4. **How sensitive is the threshold $w(D,X)$ to the hyperparameter $k$ in $\mathcal{T}(D,X)$?** [Q3]
>
> We clarify that the the threshold $w(D,X)$ is insensitive to the choice of the hyperparameter $k$ and illustrated this point by adding experiments. For the complete experimental data and explanation, please kindly refer to our detailed response to Reviewer Lzru (Q2).

---

> > ### Author Rebuttal · Reviewer_H4sy · 2026-04-01
> >
> > Thank you for your answer.

---

> > > ### Author Response · Authors · 2026-04-01
> > >
> > > Thank you for raising the score. We are pleased that our response addressed your concerns, which also improves the quality of this work. Once again, we appreciate your positive and valuable feedback.

---

### Official Review · Reviewer_Lzru · 2026-03-10

**Soundness:** 4
**Presentation:** 3
**Significance:** 3
**Originality:** 3
**Overall Recommendation:** 5
**Confidence:** 3

**Summary:**

The author proposed ST-BCP, which uses non-conformity score transformation to improve coverage gap occurred in backward conformal prediction.

**Compliance With Llm Reviewing Policy:**

Affirmed.

**Final Justification:**

The authors propose ST-BCP, a meaningful contribution to conformal prediction that addresses the coverage gap arising in backward conformal prediction. The paper is theoretically solid, and the experiments are well designed and provide strong support for the authors’ claims. The authors also addressed the questions raised during the rebuttal by adding helpful details and clarifications. Overall, I believe this paper would be a valuable contribution to the community and merits acceptance.

**Key Questions For Authors:**

- Interpretation / discussion for some regularity conditions

Theorem 3.2 requires regularity condition specified in the corresponding appendix.

$$
\lim_{n \to \infty}
\left(
\frac{1}{n}\sum_{i=1}^n h_i-\frac{1}{n}\sum_{i=1}^n h_i^y
\right)
= 0
$$

Could the authors clarify the implication of this? Technically, does it mean that adding an arbitrary test point has an asymptotically negligible effect on \( h_i \) as the calibration set size tends to infinity? Additionally, if the transformation is independent of \( D \), then Remark 3.3 holds trivially, since \( h_i = h_i^y \), as the authors noted. However, are there other settings in which Remark 3.3 holds besides the case where the transformation is independent of \( D \)? A further discussion of this point would be very helpful for the audience.


- Size constraint rule

I think the authors did not specify the value of \(k\) used in the kNN-based size-constraint rule. Reporting this choice, along with results for varying values of \(k\), would be helpful for comparing the effect of matching \(k\) against using a constant \(\mathcal{T}\).

**Limitations:**

See key questions.

**Strengths And Weaknesses:**

- Soundness

Theoretically solid paper with clear motivation grounded in the previous work [1].


- Presentation

The paper is well structured, and the figures and tables effectively convey the authors’ intent.


- Significance

Although this work extends [1] and may therefore inherit some of its limitations—particularly the difficulty of controlling the desired coverage—it explores an important and worthwhile direction in conformal prediction.

- Originality

The work has novelty with a clear motivation stemming from the reliance on Markov’s inequality in the previous work [1].

[1] Backward Conformal Prediction, Gauthier et al., 2025

---

> ### Author Rebuttal · Authors · 2026-03-31
>
> Thank you for the valuable comments. Please find our response below.
>
> 1. **Discussion about P4: $\underset{n\rightarrow\infty}{\lim}\left(\frac{\sum_{i=1}^n{h_i}}{n}-\frac{\sum_{i=1}^n{h_i^y}}{n}\right)=0$** [Q1]
>
> Yes, your understanding is right. This condition implies that the addition of a single pseudo-test point has an asymptotically negligible impact on the average of the transformed calibration scores. Since the transformation typically depends on the statistics of the input dataset $D$ and feature $X$, the perturbation caused by a single point within $D$ becomes negligible as the size of the calibration set grows sufficiently large.
>
> 2. **Other settings in which $h_i = h_i^y$ holds** [Q1]
>
> The condition $h_i = h_i^y$ holds if the transformation depends only on highly localized statistics (e.g., nearest neighbors within a fixed radius) and the pseudo-test point does not fall within the local neighborhood of any calibration points. Furthermore, if the underlying statistics are sufficiently robust (e.g., quantiles) and the addition of a pseudo-test point does not shift these quantiles, then $h_i = h_i^y$ will also remain valid.
>
> 3. **Analysis of the k-NN parameter $k$** [Q2]
>
> We apologize for not reporting the hyperparameter $k$ used in the k-NN. In Table 1, the value of $k$ for $\mathcal{T}(D,X)$ was set to 20. We conducted experiments using $k \in \textbraceleft\ 20,21,30,70 \textbraceright$ under a fixed calibration set and test point to report the distribution of adaptive $\mathcal{T}(D,X)$ under different $k$ and the sensitivity of $w(D,X)$ and $T(D,X)$ to $k$.
>
> |  | $\mathcal{T}=1$ | $\mathcal{T}=2$ |$\mathcal{T}=3$ |
> | -------- | -------- | -------- | -------- |
> |  $k$=20    | 0.68     | 0.31     | 0.01     |
> | $k$=21     | 0.63     | 0.36     | 0.01     |
> | $k$=30     | 0.55     | 0.45     | 0     |
> | $k$=70     | 0.67     | 0.33     | 0     |
>
> Secondly, to assess the sensitivity of $\mathcal{T}(D,X)$ and $w(D,X)$ to $k$, we calculated the absolute variation distribution of $\mathcal{T}(D,X)$ and $w(D,X)$ at the calibration set and test points relative to the baseline of $k=20$ :
>
> |  | $\vert\Delta \mathcal{T}\vert=0$ | $\vert\Delta \mathcal{T}\vert=1$ | $\vert\Delta \mathcal{T}\vert=2$ |
> | -------- | -------- | -------- | -------- |
> | $k=21$     | 0.92     | 0.085     | 0     |
> | $k=30$     | 0.74     | 0.26     | 0     |
> | $k=70$     | 0.67     | 0.32     | 0.005     |
>
> |  | $\vert\Delta w\vert=0$ | $0<\vert\Delta w\vert\leq0.5$ | $0.5<\vert\Delta w\vert\leq1$ |$1<\vert\Delta w\vert$ |
> | -------- | -------- | -------- | -------- | -------- |
> | $k=21$     | 0.92     | 0.069     | 0.005     | 0.009     |
> | $k=30$     | 0.74     | 0.18     | 0.020     | 0.059     |
> | $k=70$     | 0.67     | 0.21     | 0.045     | 0.075     |
>
> Even when $k$ is more than tripled (from 20 to 70), most $\mathcal{T}(D,X)$ remains unchanged. When a shift does occur, Its absolute value of change is almost only 1. Because $\mathcal{T}(D,X)$ is stable, the resulting threshold $w(D,X)$ is highly robust. Deviations in $w(D,X)$ are predominantly zero or extremely marginal.

---

> > ### Author Rebuttal · Reviewer_Lzru · 2026-04-02
> >
> > Thank you for the rebuttal. I look forward to seeing a revised version that incorporates the additional information provided in the rebuttal. I have updated my score accordingly.

---

> > > ### Author Response · Authors · 2026-04-03
> > >
> > > Thank you for reviewing our response and raising your score. We will incorporate the detailed interpretation of the regularity condition $P4$, as well as the sensitivity analysis of the hyperparameter $k$ in the kNN-based size-constraint rule, into the final version. We sincerely appreciate your time and effort in reviewing our work.

---

### Official Review · Reviewer_PLDF · 2026-03-13

**Soundness:** 3
**Presentation:** 3
**Significance:** 2
**Originality:** 3
**Overall Recommendation:** 4
**Confidence:** 4

**Summary:**

This paper studies Backward Conformal Prediction (BCP) under explicit prediction-set size constraints. The main motivation is that the coverage bound used in standard BCP is often overly conservative, creating a substantial gap between the estimated guarantee and the empirical miscoverage. To address this, the authors propose ST-BCP, which applies a data-dependent transformation of the non-conformity score so that the transformed score distribution is better suited to the Markov-inequality-based analysis used in BCP. The method is designed with a symmetric parameterization intended to preserve exchangeability, and the paper further develops a leave-one-out estimator, an invariance analysis for the resulting prediction sets, and a theoretically motivated monotone transformation family with a computable approximation $I_w$. Experiments on CIFAR-10, CIFAR-100, and Tiny-ImageNet show that ST-BCP consistently reduces the gap between the estimated coverage bound and empirical miscoverage, while leaving empirical miscoverage itself nearly unchanged.

**Compliance With Llm Reviewing Policy:**

Affirmed.

**Final Justification:**

The responses address my concerns, and I have raised my score.

**Key Questions For Authors:**

See above

**Limitations:**

See above

**Strengths And Weaknesses:**

Strengths:
1. The paper addresses a clear and practically relevant weakness of BCP, namely the looseness of the estimated coverage bound. This is a well-motivated problem because an excessively conservative bound reduces the practical usefulness of the guarantee.
2. The core idea is novel, instead of redesigning the calibration procedure, the paper tightens the bound through a data-dependent transformation of the non-conformity score.

Weaknesses:
1. Several theoretical results are stated under appropriate conditions or similarly vague regularity assumptions, but these conditions are neither especially transparent nor obviously mild. Some assumptions used for the leave-one-out consistency and optimal-transformation analyses seem fairly strong and mainly proof-enabling. The paper does not adequately discuss whether these assumptions are realistic in modern deep classification settings.
2. The experiments mainly show tighter estimated bounds, not stronger practical utility. Since empirical miscoverage is nearly unchanged, the paper does not clearly demonstrate why the improvement matters in downstream decision-making.


Questions:
1. Given the failure mode discussed in the low-coverage regime, why are the robust transformation and corrected coverage level not included as main baselines?
2. Can the authors provide a concrete downstream example where a tighter estimated bound leads to a materially different decision?
3. Appendix C discusses the regression setting, but the main paper only reports classification experiments. Have the authors attempted any non-classification tasks, or can they clarify why the current method is primarily evaluated in classification?

Limitations:
1. The limitations are generally well acknowledged, but they could be made more concrete. In particular, the method appears less stable in the low-coverage regime, the theory depends on fairly strong assumptions, the experiments are limited mostly to image classification, and the practical value of tighter estimated bounds could be clarified further.

---

> ### Author Rebuttal · Authors · 2026-03-31
>
> Thank you for the valuable comments. Please find our response below.
>
> 1. **Assumptions under modern deep models** [W1]
>
> We clarify that the theoretical assumptions in this paper do not impose restrictions on the architectures of modern deep learning models themselves. In deep classification tasks, as long as the non-conformity scores provided by the model on the calibration set are independent and identically distributed (i.i.d.), the "black-box" nature of the model does not affect the validity of our theory. The conditions and assumptions we define primarily constrain the design of the size constraint rule $\mathcal{T}(D,X)$ and the score transformation $h(s;D,X)$.
>
> 2. **The impact on downstream decision-making** [W2, Q2]
>
> A tighter coverage lower bound can lead to substantially different downstream decisions, offering significant practical value in risk-sensitive tasks.
>
> Consider a medical diagnostic system where routine cases are automated if the estimated coverage lower bound exceeds a safety threshold; otherwise, it conservatively defers to a doctor. With standard BCP, the estimated coverage lower bound may fall severely below the threshold even if the true coverage is safe. This over-conservatism forces safe cases into manual review. By applying ST-BCP, the estimated coverage lower bound successfully exceeds the required threshold, accurately reflecting the system's true reliability and significantly reducing unnecessary human intervention.
>
> 3. **Why are Robust Transformation and Corrected Coverage not primary baselines?** [Q1]
>
> They were not included as primary baselines because, despite relieving the problem that Taylor approximation is seriously invalid, they lack the rigorous theoretical foundation. Specifically:
>
> * Robust transformation: While intended to reduce the Taylor error, it cannot guarantee that it is superior to or equal to the identity transformation (i.e. BCP) in reducing the Markov error.
> * Corrected Coverage Level: This approach utilizes a different probability inequality than BCP that will produce different error.  More importantly, it currently lacks a rigorous proof demonstrating consistent convergence for the LOO estimator.
>
> Therefore, we have placed these methods in the appendix as supplementary strategies for extreme cases, rather than as core baselines.
>
>
> 4. **Why is ST-BCP primarily evaluated in classification?** [Q3]
>
> We mainly evaluate our method in classification tasks, as extending ST-BCP to a continuous regression setting poses fundamental computational challenges that are beyond the scope of this paper. In ST-BCP, it is necessary to calculate the exact size of the prediction set at the given miscoverage level $\alpha$. When the transformation $h$ depends on the calibration set, to maintain strict exchangeability, our method couples the candidate label $y$ into the transformation. At this point, the prediction set is:
>
> $$
> C_h^{\alpha}(X_{n+1})=\textbraceleft\ y \in \mathcal{Y} :  \frac{(n+1)h_{n+1}(y)}{\sum_{i=1}^{n} h_i^y + h_{n+1}(y)} < \frac{1}{\alpha}  \textbraceright
> $$
>
> For classification, we can effectively calculate the exact size of this prediction set by traversing over a limited discrete label space. However, for regression (where $\mathcal{Y}$ is continuous), calculating the exact size of this prediction set becomes difficult. Therefore, we did not apply the method to regression. The discussion in Appendix C is merely to explore the theoretical possibilities for the future.

---

> > ### Author Rebuttal · Reviewer_PLDF · 2026-04-01
> >
> > The responses address my concerns, and I have raised my score.

---

> > > ### Author Response · Authors · 2026-04-02
> > >
> > > Thank you for taking the time to read our rebuttal and for your positive feedback! We are excited to hear that our responses have resolved your concerns about this work. We noticed your kind note mentioning that you have raised your score. However, it appears the OpenReview system hasn't reflected this update yet. Could you please kindly check if the score change was successfully saved? It may be edited in Official Review.

---

### Official Review · Reviewer_mAks · 2026-03-16

**Soundness:** 3
**Presentation:** 3
**Significance:** 2
**Originality:** 3
**Overall Recommendation:** 4
**Confidence:** 3

**Summary:**

The paper studies the looseness of the coverage certificate in backward conformal prediction (BCP), where a fixed upper bound on the prediction set size is specified and the goal is to estimate the corresponding coverage guarantee. The authors attribute the gap between the estimated coverage bound and the empirical miscoverage to the looseness of the Markov inequality used in BCP. To address this issue, they propose applying a symmetric transformation to the nonconformity scores before constructing the e-variable, to reshape the score distribution so that the Markov bound becomes tighter. The paper analyzes a class of such transformations and derives a practical step-function transformation that can be computed from the data. Empirical results on several image classification datasets show that the proposed method reduces the gap between the leave-one-out coverage estimate and the empirical miscoverage while preserving the validity guarantees of BCP under fixed set-size constraints.

**Compliance With Llm Reviewing Policy:**

Affirmed.

**Final Justification:**

The paper addresses a meaningful problem in backward conformal prediction: overly conservative coverage certification under fixed set-size constraints. The proposed transformation is simple and appears technically sound. The empirical results are consistent, and the additional rebuttal experiments on different nonconformity scores increased my confidence in the method.
Some of my earlier theoretical questions are only partially resolved, especially about the full mechanism behind the reduction in the final estimation gap. Still, the rebuttal clarified the motivation and strengthened the empirical case enough to positively change my assessment.

Overall, I now view the paper as a useful contribution within the BCP setting and lean borderline accept.

**Key Questions For Authors:**

1. The paper argues that the transformation improves the tightness of the coverage certificate by reshaping the score distribution so that the Markov inequality becomes tighter. However, the full BCP pipeline involves several steps (score transformation $\rightarrow$ e-variable construction $\rightarrow$  $\tilde \alpha(h)$ $\rightarrow$  LOO estimation). Can the authors formally show that the proposed score transformation provably reduces the coverage estimation gap in this full pipeline, rather than only improving the tightness of the Markov bound at the score level?

2. The proposed step-function transformation effectively performs a thresholding of the scores. From an e-value perspective, this can be interpreted as post-hoc reshaping the e-variable distribution to better match the tight case of the Markov inequality. How should this operation be interpreted statistically? In particular, does this reshaping affect the evidential meaning or optimality properties of the original e-variable?

3. The theoretical optimal transformation derived in the paper depends on unknown conditional probabilities, while the implemented step-function approximation is only motivated by this result. Can the authors clarify whether the practical transformation is solving a well-defined approximation of the original optimization problem, or whether it should be viewed as a heuristic inspired by the theoretical analysis?

4. Theorem 3.2 shows that the prediction sets remain unchanged under the considered transformation class. Given this invariance, the improvement comes entirely from the coverage certificate. How should practitioners interpret this improvement? Does reducing the gap between the estimated bound and empirical miscoverage correspond to a strictly better certification procedure in the BCP framework?

5. The analysis relies on a first-order Taylor approximation in deriving the coverage bound. To what extent does the observed gap in practice come from the looseness of the Markov inequality versus the Taylor approximation step? Can the authors empirically or theoretically separate these two sources of error?

6. The experiments are conducted with a fixed nonconformity score derived from the classifier outputs. Since the proposed method reshapes the score distribution to improve the Markov bound, how sensitive is the method to the choice of nonconformity score? Do the improvements persist under different score functions commonly used in conformal prediction (e.g., APS/RAPS scores)?

**Limitations:**

Yes

**Strengths And Weaknesses:**

**Strengths**

The paper studies the looseness of the coverage certificate in backward conformal prediction (BCP). It proposes a simple score transformation that can be applied without retraining the model and preserves the validity guarantees of BCP. The paper provides theoretical analysis of the transformation framework and studies the consistency of the leave-one-out estimator used for coverage certification. Experiments on several image classification datasets show that the proposed method reduces the gap between the estimated coverage bound and the empirical miscoverage.

**Weaknesses**

The proposed method does not change the prediction sets and mainly affects the numerical behavior of the coverage certificate. As a result, the practical impact is limited to improving the tightness of the reported coverage bound. In addition, the central explanation based on tightening the Markov inequality is largely heuristic, and the paper does not formally establish the link between the score transformation and the reduction of the coverage estimation gap in the full BCP procedure. The theoretically optimal transformation depends on unknown quantities, and the practical step-function approximation is only loosely connected to the theoretical formulation. The experiments focus on the gap between the estimated bound and empirical miscoverage, which mainly demonstrates improved calibration of the certificate rather than broader practical benefits.

---

> ### Author Rebuttal · Authors · 2026-03-31
>
> Thank you for the valuable comments. Please find our response below.
>
> 1. **Practical value of tighter coverage lower bound** [W1,Q4]
>
> We clarify that the goal of backward conformal prediction (BCP) is to precisely estimate the coverage rate with a specific upper bound of prediction set size. For this purpose, our method delivers a significantly tighter lower bound on coverage compared to BCP, thereby reducing the conservatism inherent in coverage estimation. This improvement is pratically valuable in downstream applications, as excessively conservative coverage bounds may prompt unwarranted human intervention. For instance, in medical diagnostic systems, an estimated coverage rate falling below stringent safety thresholds could result in unnecessary case referrals to human doctors. To improve the clarity, we will incorporate  the description of pratical value in the final version.
>
>
> 2. **Theoretical effectiveness of our method** [W2,Q1]
>
> First, the gap between the estimated coverage of BCP and the true coverage can be decomposed into two parts: $P_{true}-P_{estimate}=(P_{true}-P_{lower})+(P_{lower}-P_{estimate}),$ where $P_{true}, P_{estimate}, P_{lower}$ denote the true coverage, the estimated value, and the theoretical lower bound of the coverage, respectively.
> Theoretically, our method reduces the gap in the first part ($P_{true}-P_{lower}$) by providing a tighter coverage bound (Corollary 3.6). Besides, we formally show that the estimated coverage of our method can converge to the lower bound through the LOO estimator (Theorem 3.1). Empirically, the experimental results in Table 1 and Figures 2-3 validate that our method significantly narrows the estimation gap of coverage. We believe the clarification can provide a comprehensive understanding of our method for readers.
>
>
> 3. **Clarification of score transformations** [W3,Q3]
>
> We clarify that the proposed transformation $\mathbb{I}_w$ is a rigorous method with solid theoretical backing, rather than a mere heuristic.
> In particular, we construct $\mathbb{I}_w$ by applying an idempotent lifting operator $G$ to the identity transformation that corresponds to the original BCP. Theoretically, we prove that the operator $G$ provably reduces the gap between the lower bound and the true coverage (Corollary 3.6). Moreover, the resulting transformation preserves the step-wise structure of the optimal transformation, thereby exerting a similar influence on the lower bound.
>
>
> 4. **E-value interpretation of step-wise transformation** [Q2]
>
> The proposed transformation does not compromise the e-variable property. This is because we adopt a symmetric parametrization that preserves the exchangeability of the transformed scores, thereby ensuring that $E^{n+1}(Y_{n+1}, h)$ remains a valid e-value. Statistically, this reshaping can be interpreted as leveraging prior information in the size constraint rules to derive a sharper probability inequality, while fully respecting the definition of an e-value.
>
>
> 5. **Separating Taylor and Markov errors** [Q5]
>
> We formally separate these two errors through the following formula:
>
> $$1 \underset{\text{Markov Error}}{\geq} \mathbb{E}\left[\frac{\mathbb{P}\left(Y\notin C\left(X\right)\mid\tilde{\alpha}\right)}{\tilde{\alpha}}\right] \underset{\text{Taylor Error}}{\approx}\frac{\mathbb{E}\left[\mathbb{P}\left(Y\notin C\left(X\right)\mid\tilde{\alpha}\right)\right]}{\mathbb{E}[\tilde{\alpha}]}$$
> Intuitively, the Markov and Taylor errors originate from distinct stages, i.e., scaling and approximation. Besides, we provide a discussion whether our transformation affects the Taylor error in our response to Reviewer H4sy (Q2).
>
>
> 6. **The improvement effects of different score functions**[Q6]
>
> Our method is highly insensitive to the choice of non-conformity score function. We conducted experiments using various non-conformity score functions on CIFAR-10 with ResNet50 ($\mathcal{T}=2, n=200$).
>
> |**BCP / ST-BCP (ours)** |  | | | |
> | :--- | :---: | :---: | :---: | :---: |
> | **Score Function** | **MisCov(%)** | **STD(%) ↓** | **MSE(×$10^4$) ↓** | **GAP(%) ↓** |
> | APS | 2.80 / 2.80 | 2.02 / **1.07** | 4.07 / **1.14** | 48.2 / **0.84** |
> | Cross-Entropy | 2.26 / 2.26 | 1.28 / **0.89** | 1.63 / **0.79** | 5.38 / **0.72** |
> | Rank | 2.20 / 2.20 | 1.34 / **1.12** | 1.79 / **1.26** | 35.6 / **1.14** |
> | $1-\pi_y(X)$ | 2.58 / 2.58 | 1.64 / **1.09** | 2.69 / **1.19** | 13.7 / **0.92** |
>
> From the results, we observe that ST-BCP consistently reduces the coverage gap (GAP), enhances the stability of LOO estimator. The improvement is particularly substantial when the baseline score performs poorly. For example, ST-BCP drastically reduces the GAP of the APS score from 48.2% to 0.84%.

---

> > ### Author Rebuttal · Reviewer_mAks · 2026-04-02
> >
> > Thank you for the clarifications. The rebuttal improves the presentation and adds useful experiments on different nonconformity scores.
> >
> > That said, my main concern is only partially resolved. The rebuttal explains why the method can tighten the lower bound, but it still does not fully establish why this should reduce the final estimation gap in the full BCP pipeline, which also involves the data-dependent $\tilde \alpha$ and the LOO estimator.
> >
> > The discussion of e-values also clarifies that validity is preserved, but this is not the same as clarifying the statistical role of the transformation. In particular, it remains unclear whether the method should be viewed as improving certification itself or as reshaping the e-variable to better match the Markov inequality used in BCP.
> >
> > Overall, the rebuttal is helpful, but I think the main theoretical link could still be clarified further.

---

> > > ### Author Response · Authors · 2026-04-03
> > >
> > > Thank you for the detailed feedback.
> > >
> > > 1. **Why can ST-BCP reduce the final estimation gap in the full BCP pipeline?**
> > >
> > > To more clearly illustrate how the score transformation reduces the final estimation gap, we decompose this gap via the following inequality:
> > >
> > > $$
> > > \lvert\mathbb{P}(Y\in C(X))-(1-\hat{\alpha}^{LOO})\rvert\leq\vert\mathbb{P}(Y\in C(X))-(1-\mathbb{E}[\tilde{\alpha}])\rvert+\lvert\hat{\alpha}^{LOO}-\mathbb{E}[\tilde{\alpha}]\rvert
> > > $$
> > >
> > > This demonstrates that the final estimation gap is bounded by two components: the coverage lower bound gap and the LOO estimation gap.
> > >
> > > * Coverage lower bound gap: We proved that the transformation $\mathbb{I}_w$, derived by applying the idempotent lifting operator $G$ to the identity transformation (BCP), reduces $\lvert\mathbb{P}(Y\in C(X))-(1-\mathbb{E}[\tilde{\alpha}])\rvert$ when the Taylor approximation holds (see Corollary 3.6).
> > > * LOO estimation gap: The reduction of this term is supported by our empirical results. As shown in Table 1 and Figure 2, we compared the $MSE(\hat{\alpha}^{LOO},\mathbb{E}[\tilde{\alpha}])$ under the identity transformation and $\mathbb{I}_w$. The results consistently show a reduced MSE, providing empirical evidence that $\lvert\hat{\alpha}^{LOO}-\mathbb{E}[\tilde{\alpha}]\rvert$ is smaller in practice.
> > >
> > > Therefore, our method provides both theoretical and empirical evidence that the final estimation gap in the full pipeline is substantially reduced in practice. This phenomenon can be observed intuitively in the KDE plot of $\hat{\alpha}^{LOO}$ in Figure 1. Compared to the identity transformation, $\mathbb{I}_w$ shifts the LOO distribution closer to the true miscoverage (reducing the coverage lower bound gap) and makes it more concentrated around the corresponding miscoverage level expectation (reducing the LOO estimation gap).
> > >
> > > 2. **How should ST-BCP be viewed?**
> > >
> > > We clarify that the method should be viewed as improving certification itself. This is because our proposed transformation $\mathbb{I}_w$ is rigorously derived by optimizing the difference between the true coverage and the coverage lower bound as the objective functional. As for "reshaping the e-variable distribution to better match the Markov inequality used in BCP," this is actually our interpretation of the transformation $\mathbb{I}_w$ from a probability theory perspective. It intuitively explains why our score transformation works.

---

### Decision · Program_Chairs · 2026-04-30

**Decision:**

Accept (regular)

**Comment:**

The authors address a well-known limitation in Backward Conformal Prediction (BCP), the overly conservative coverage bounds resulting from the looseness of Markov's inequality. By introducing ST-BCP, a data-dependent and symmetric transformation of non-conformity scores, the authors provide a good solution that significantly tightens the coverage bound while strictly preserving exchangeability and validity guarantees.

The reviewers uniformly praised the paper's theoretical originality and the strong empirical results, noting that the reduction of the coverage gap is substantial and consistent across benchmarks. During the review process, some valid questions were raised regarding the practical utility of merely tightening the bound without changing the prediction sets, the computational overhead for large label spaces, the separation of Markov and Taylor approximation errors, and the robustness of the method to hyperparameters in the size-constraint rules.

The authors provided an exceptionally thorough and convincing rebuttal that successfully resolved these concerns. They clearly articulated the practical value of tighter certification in risk-sensitive domains (such as medical diagnostics), where overly conservative bounds frequently trigger unwarranted human intervention. Furthermore, the authors provided a rigorous theoretical decomposition proving that their transformation strictly improves the Markov bound without exacerbating the Taylor approximation error. The addition of the wall-clock time analysis and the sensitivity analysis for the $k$-NN parameter further strengthened the paper's practical standing.

Overall, this is a technically solid and highly original paper that advances the field of uncertainty quantification. I recommend accepting this paper.